# LoSplit: Loss-Guided Dynamic Split for Training-Time Defense Against Graph Backdoor Attacks

**Di Jin**[1]   **Yuxiang Zhang**[1]   **Bingdao Feng**[1][*]   **Xiaobao Wang**[1]   **Dongxiao He**[1]   **Zhen Wang**[2]

[1]College of Intelligence and Computing, Tianjin University, Tianjin, China
[2]School of Cybersecurity, Northwestern Polytechnical University, Xi'an, Shaanxi, China
`{jindi,yzhang4062,fengbingdao,wangxiaobao,hedongxiao}@tju.edu.cn`
`w-zhen@nwpu.edu.cn`

## Abstract

Graph Neural Networks (GNNs) are vulnerable to backdoor attacks. Existing defenses primarily rely on detecting structural anomalies, distributional outliers, or perturbation-induced prediction instability, which struggle to handle the more subtle, feature-based attacks that do not introduce obvious topological changes. Our empirical analysis reveals that both structure-based and feature-based attacks not only cause early loss convergence of target nodes but also induce a class-coherent loss drift, where this early convergence gradually spreads to nearby clean nodes, leading to significant distribution overlap. To address this issue, we propose *LoSplit*, the first training-time defense framework in graph that leverages this early-stage loss drift to accurately split target nodes. Our method dynamically selects epochs with maximal loss divergence, clusters target nodes via Gaussian Mixture Models (GMM), and applies a Decoupling-Forgetting strategy to break the association between target nodes and malicious label. Extensive experiments on multiple real-world datasets demonstrate the effectiveness of our approach, significantly reducing attack success rates while maintaining high clean accuracy across diverse backdoor attack strategies. Our code is available at: github.com/zyx924768045/LoSplit.

## 1 Introduction

Graphs serve as a ubiquitous form of data structure [1] in many real-world applications such as social networks [2], recommendation systems [3], and financial applications [4], where nodes represent entities and edges capture their relationships. Graph Neural Networks (GNNs) [5, 6, 7] effectively capture graph information via iterative message passing, and have shown strong performance in tasks like node classification [8].

However, GNNs have been shown to be susceptible to backdoor attacks [9, 10, 11, 12, 13, 14, 15, 16], where adversaries embed triggers (either predefined or adaptively generated) in the form of nodes, subgraphs or feature perturbations, into training data to create shortcuts between target nodes and malicious labels. Due to the message-passing mechanism, these triggers will deceive GNNs to misclassify trigger attached nodes into malicious label during inference, while preserving high performance on clean nodes, thereby posing serious risks in safety-critical applications.

Existing defense against such attack adopt various strategies to detect and mitigate backdoor impact. Prune [11] drops edges that link nodes that exhibit low cosine similarity. OD [12] identifies outlier nodes based on distributional differences and removes them to neutralize trigger influence. RIGBD [17] observes that simple pruning and outlier detection methods can be easily bypassed by carefully crafted triggers, and thus introduces a combination of randomized edge dropping and robust training to weaken trigger influence and restore model integrity. However, defending against more subtle, feature-based backdoor attacks remains a significant challenge. Unlike structural perturbations

---

[*]Corresponding author.

39th Conference on Neural Information Processing Systems (NeurIPS 2025).

that introduce easily detectable topological anomalies, feature-based triggers operate by subtly manipulating node attributes, often without perturbing the overall graph structure. As highlighted by SPEAR [13], these attacks can preserve the original topology while selectively modifying high importance features, effectively bypassing common graph anomaly detectors. This makes existing defenses that rely on detecting abrupt structural deviations or outliers ineffective, highlighting the need for more comprehensive training-time defenses.

To address this, we hypothesize that these two types of attacks, despite their differences, share certain early training traits. This is consistent with our observation that GNNs often exhibit shortcut learning behaviors in which trigger-embedded nodes rapidly fitting into malicious labels. Similar phenomenon has been reported in deep neural networks (DNNs) [18], where poisoned samples show a sharp and early drop in loss values. This occurs because backdoor triggers introduce strong signals that models latch onto early in training, causing target nodes to converge significantly faster than clean nodes.

However, directly applying this intuition to graphs presents unique challenges. In images, each poisoned sample is an independent image (analogous to a graph) with a strong localized trigger (e.g., a black pixel), which the model quickly fits, leading to a stable and fast loss convergence (see Appendix L). In contrast, graph backdoor attacks on node classification task conducts over a single interconnected graph, where poisoned samples are only a few nodes attached by triggers. Different from images, triggers in graphs does not influence the graph directly. Instead, it propagates **step-by-step** through message passing across limited number of hops to its neighbors, which makes the loss behavior unstable as training epoch progresses. As shown in Fig. 1, we observe a previously unexplored phenomenon we call *class-wide loss drift*: loss distributions of nodes fluctuate, generally shifting leftward but occasionally reversing, with frequent overlaps across classes. This instability renders fixed ratios and global thresholds typically used in images ineffective in graphs, thus posing a distinct challenge:

*How to precisely identify target nodes even in the presence of unstable class-wide loss drift?*

To address this, we propose **Lo**ss-guided dynamic target node **Split** framework (*LoSplit*), the first training-time defense in graphs that introduces several key innovations. First, it identifies malicious label by computing the intra-class loss variance, then it performs **step-by-step** optimal split epoch selection using intercluster divergence, rather than relying on a fixed heuristic early epoch like mainstream image approaches do. Second, it applies Gaussian Mixture Models (GMM) clustering to adaptively adjust splitting thresholds to split target nodes and clean nodes. Finally, with separated target and clean nodes, it restores the backdoored model using a Decoupling-Forgetting strategy. For target nodes, we apply random label reassignment combined with gradient ascent to maximally de-

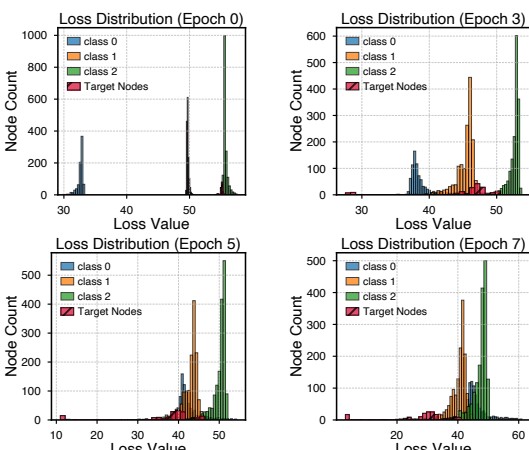

Figure 1: Class-wide loss drift phenomenon across epochs.

couple the shortcut association between them and the malicious label. For clean nodes, we train normally to maintain model performance. Our main contributions are threefold:

**(1)** We propose *LoSplit*, the first training-time graph backdoor defense. **(2)** We design a novel Early-Stage Dynamic Split for target nodes identification and Decoupling-Forgetting for Backdoor Recovery. **(3)** Extensive experiments on real-world datasets demonstrate that *LoSplit* consistently reduces attack success rates while preserving clean accuracy across diverse backdoor attacks.

## 2 Related Works

### 2.1 Graph Backdoor Attacks

Early methods (e.g., SBA [9], GTA [10]) use fixed subgraph patterns, while UGBA [11] and DPGBA [12] improve stealthiness by exploiting homophily and distributional priors. SPEAR [13] further advances this by perturbing node features only. More details are provided in Appendix A.1.

### 2.2 Graph Backdoor Defense

Graph Backdoor defense methods fall into two categories. On is Detection-Deletion type of method (e.g., Prune [11], OD [12]) and the other is Detection-Robust Training method such as RIGBD [17]. While effective in structural perturbation attacks, they fail to counter more stealthy feature perturbation attacks like SPEAR [13] due to the absence of topological anomalies. More details are provided in Appendix A.2.

### 2.3 Training-time Backdoor Defense

Training-time backdoor defenses, initially developed in the image domain such as ABL [18], DBD [19], ASD [20], PIPD [21], and HARVEY [22], identify and mitigate poisoned samples by progressively isolating them based on early training behaviors. Although effective in vision, these methods overlook graph-specific properties such as message passing, neighborhood aggregation, and structural homophily, making them less effective or inapplicable to graphs. More details are provided in Appendix A.3.

## 3 Preliminaries

**Graph Representation Learning**. We define an undirected graph as $\mathcal{G} = (\mathcal{V}, \mathcal{A}, \mathcal{E}, \mathcal{X}, \mathcal{Y})$, where $\mathcal{V} = \{v_1, \ldots, v_N\}$ is the set of $N$ nodes and $\mathcal{A} \in \mathbb{R}^{N \times N}$ is the adjacency matrix denoting the relationship between nodes: $\mathcal{A}_{ij} = 1$ if node $v_i$ is connected to node $v_j$, $\mathcal{A}_{ij} = 0$ otherwise. $\mathcal{E} \subseteq \mathcal{V} \times \mathcal{V}$ is the set of edges, and $\mathcal{X} \in \mathbb{R}^{N \times d}$ represents the node features. Each node $v_i \in \mathcal{V}$ is associated with a feature vector $x_i$ and a ground-truth label $y_i \in \mathcal{Y}$. Graph Convolutional Networks (GCNs) [5] compute node embeddings through spectral graph convolution using the layer-wise propagation rule:

$$\mathbf{H}^{(l+1)} = \sigma\left(\hat{\mathbf{A}}\mathbf{H}^{(l)}\mathbf{W}^{(l)}\right), \tag{1}$$

where $\mathbf{H}^{(l)} \in \mathbb{R}^{N \times h_l}$ contains node embeddings at layer $l$ (initialized with $\mathbf{H}^{(0)} = \mathbf{X}$), $\hat{\mathbf{A}} = \tilde{\mathbf{D}}^{-1/2}\tilde{\mathbf{A}}\tilde{\mathbf{D}}^{-1/2}$ represents the normalized adjacency matrix with added self-loops ($\tilde{\mathbf{A}} = \mathbf{A} + \mathbf{I}_N$), $\tilde{\mathbf{D}}$ is the corresponding degree matrix, $\mathbf{W}^{(l)} \in \mathbb{R}^{h_l \times h_{l+1}}$ denotes trainable parameters, and $\sigma(\cdot)$ is an activation function such as ReLU.

**Graph Backdoor Attack**. Adversaries strategically implant stealthy trigger patterns typically in the form of subgraphs into a small number of training nodes to mislead GNNs to produce attacker-specified outputs when trigger appears but behave normally on clean nodes during inference. In other words, the model is "backdoored". A sound backdoor attack should maintain high clean accuracy (CA) and high attack success rate (ASR) simultaneously. We study *node classification task* under this type of attack, where the training graph $\mathcal{G}_T = (\mathcal{V}_T, \mathcal{A}_T, \mathcal{E}_T, \mathcal{X}_T, \mathcal{Y}_T)$ and test graph $\mathcal{G}_U = (\mathcal{V}_U, \mathcal{A}_U, \mathcal{E}_U, \mathcal{X}_U, \mathcal{Y}_U)$ are disjoint, i.e., $\mathcal{V}_T \cap \mathcal{V}_U = \emptyset$. The attacker selects a set of target (poisoned) nodes $\mathcal{V}_B \subset \mathcal{V}_T$ and attaches backdoor triggers $\mathcal{G}_j$ to nodes $v_j \in \mathcal{V}_B$, resulting in perturbed nodes $v_j \oplus \mathcal{G}_j$ and trigger-embedded graph $\mathcal{G}'_T = (\mathcal{V}'_T, \mathcal{A}'_T, \mathcal{E}'_T, \mathcal{X}'_T, \mathcal{Y}'_T)$, where $\oplus$ denotes feature or structural perturbation. These triggers may involve synthetic subgraphs (structure-based) or crafted feature shifts (feature-based). Clean nodes are denoted as $\mathcal{V}_C = \mathcal{V}_T \setminus \mathcal{V}_B$.

A backdoored model $f'$ is trained to associate triggers with a predefined malicious label $y_t$, causing target nodes misclassified into $y_t$ while maintaining correct prediction on clean nodes:

$$\begin{cases} f(v_j \oplus g_j) = y_t, & \forall v_j \in \mathcal{V}_B; \\ f(v_i) = y_i, & \forall v_i \in \mathcal{V}_C. \end{cases} \tag{2}$$

Due to message passing in GNNs, even localized triggers can propagate across neighborhoods, leading the model to learn spurious correlations between the trigger and the malicious label. This causes the model to misclassify any node containing the trigger as $y_t$ during inference, while maintaining high accuracy on clean nodes, making the attack both effective and stealthy.

**Defender's Knowledge and Goal**. Defenders can only get access to the perturbed training graph $\mathcal{G}'_T$, without the knowledge of the malicious label $y_t$, the number and position of target nodes $|\mathcal{V}_B|$, and trigger patterns $g_j$. Depending on the attack, defenders can either refine a model using $\mathcal{G}'_T$ or receive an infected model via an adversary-provided API [23]. In both cases, defenders have no access to clean validation data or ground-truth labels. Unlike previous works that assumed post-hoc access to infected models, we consider a more practical *training-time defense* setting, where target nodes are identified during model training phase. Given a trigger-embedded graph, defender's goal is to maintain high accuracy on clean nodes while decoupling the target nodes from the malicious label $y_t$. For clean nodes $v_j \in \mathcal{V}_C$, we aim to correctly predict their labels $y_j$; for target nodes $v_i \in \mathcal{V}_B$, we reduce their association with $y_t$. Specifically, the defense objective is:

$$\min_{\theta} \mathcal{L}_\theta = \sum_{v_i \in \mathcal{V}_B} \mathcal{L}(f(v_i), \tilde{y}_i) + \sum_{v_j \in \mathcal{V}_C} \mathcal{L}(f(v_j), y_j), \tag{3}$$

where $\tilde{y}_i$ is a label filter out malicious label: $\tilde{y}_i \neq y_t$ and $\mathcal{L}$ is the classic classification loss such as Cross-Entropy loss.

**Early-Stage Loss Dynamics**. We empirically investigate the early-stage loss behaviors of clean nodes ($\mathcal{V}_C$) and target nodes ($\mathcal{V}_B$) by tracking their training losses across initial epochs. To be more specific, we compare two loss functions. One is Cross-Entropy (CE), defined as $\mathcal{L}_{CE} = -\sum_{k=1}^{K} q(k|x) \log p(k|x)$. Another is Reverse Cross-Entropy (RCE): $\mathcal{L}_{RCE} = -\sum_{k=1}^{K} p(k|x) \log q(k|x)$, which has been used in noisy label learning [24] to reduce overfitting by encouraging uncertain predictions and HARVEY [22] to isolate poisoned samples. Inspired by previous work, we also adopt RCE loss to amplify the loss divergence between target nodes and clean nodes early in training, enabling more effective split.

As shown in Fig. 2a and Fig. 2b, under both CE and RCE loss, target nodes converge noticeably faster than clean nodes during the early training epochs. This phenomenon becomes significantly more pronounced under RCE loss, where target nodes rapidly approach low loss values, thus amplifying their separability from clean nodes. A detailed theoretical justification for why target nodes converge faster than clean nodes during early training and why RCE amplifies the early-stage loss divergence between target and clean nodes more effectively than CE loss is provided in Appendix C.1.

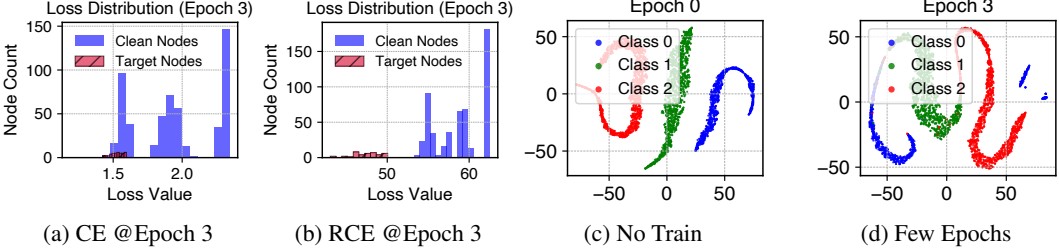

(a) CE @Epoch 3     (b) RCE @Epoch 3     (c) No Train     (d) Few Epochs

Figure 2: (a) and (b) compare the loss distributions under CE and RCE. It is evident that RCE achieves a clearer separation of target nodes compared to CE. (c) and (d) illustrate the early-stage RCE loss clustering before and after a few training epochs, where the blue cluster (corresponding to the target class) gradually splits into several sub-clusters, with the smaller sub-clusters primarily consisting of target nodes.

Another intriguing loss dynamic that we observe is that clean nodes of the same class naturally form compact loss clusters (see Fig. 2c). However, as training progresses, the cluster corresponding to the target class (blue clusters in Fig. 2d) gradually splits into two or more sub-clusters. The smaller sub-clusters predominantly consist of target nodes.

## 4 Methodology

In this section, we present the proposed defense framework *LoSplit*, whose overall framework is illustrated in Fig. 3. *LoSplit* operates in two stages: (1) Early-Stage Dynamic Split, where we

exploit the distinct early loss dynamics under RCE loss to separate target nodes from clean nodes; (2) Decoupling–Forgetting, where the identified target nodes are decoupled and forgotten from the malicious labels through a combination of random label reassignment and gradient ascent. This two-stage pipeline enables *LoSplit* to effectively split target nodes and mitigate backdoor effect without requiring access to ground-truth information. In the following, we will give the details of each stage.

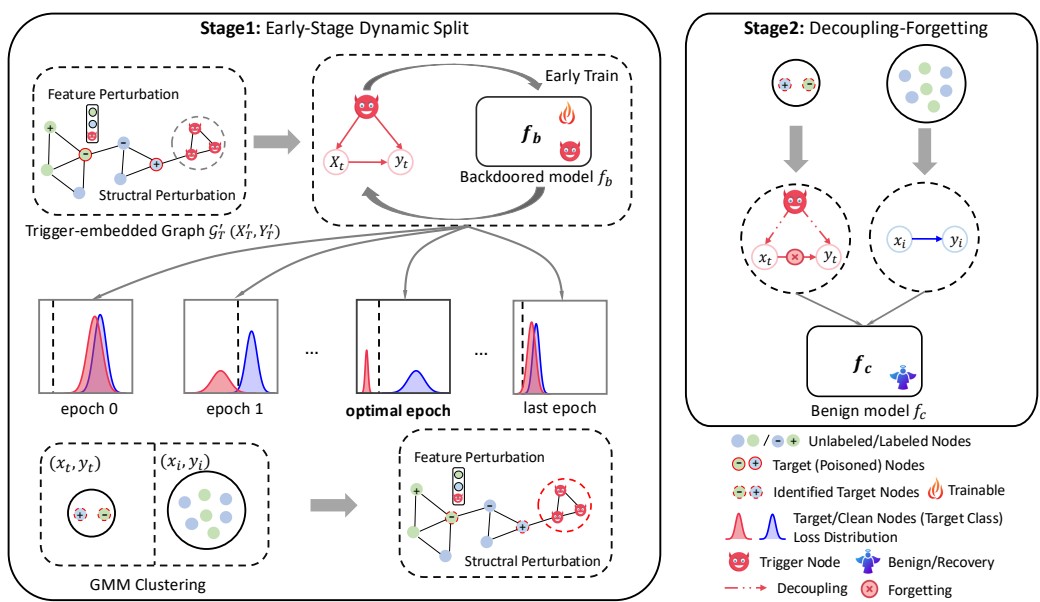

Figure 3: Overall Framework of LoSplit.

## 4.1 Target Nodes Identification via Early-Stage Dynamic Split

*LoSPlit* first leverages the loss dynamics discussed in Sec. 3 that nodes in the same malicious class tend to split into several sub-clusters to identify malicious label. We achieve this by computing the variance of the loss value of each label and select the label with the highest intra-class loss variance as malicious label:

$$y_t = \arg \max_{y_j \in \mathcal{Y}'_T} \mathrm{Var}\left( \{\ell_i^{(t)} \mid y_i = y_j\} \right), \tag{4}$$

where $\ell_i^{(t)}$ denotes the RCE loss of node $v_i$ at epoch $t$ in ascending order.

Once the malicious label $y_t$ is identified, we extract all nodes labeled $y_t$, denoted as $\mathcal{V}_{y_t}^{(t)}$. We compute their mean $\mu$ and standard deviation $\sigma$ of the RCE loss value at a given early epoch $t$. We then normalize their loss values into a standardized z-score form:

$$\zeta_i^{(t)} = \frac{\ell_i^{(t)} - \mu}{\sigma + \epsilon}, \quad \forall v_i \in \mathcal{V}_{y_t}^{(t)}, \tag{5}$$

where $\epsilon$ is a small constant to avoid division by zero.

We empirically observe that target nodes tend to form a tight cluster at the lower end of the loss spectrum. In practice, we sort the loss values within class $y_t$ in ascending order and then fit a Gaussian Mixture Model (GMM) [25] to identify the low-loss cluster (target nodes). Let $\mathcal{C}_{\mathrm{low}}^{(t)}$ and $\mathcal{C}_{\mathrm{high}}^{(t)}$ be the node cluster with smaller and larger average loss values at epoch $t$, respectively. Then we define an adaptive threshold $\tau^{(t)}$ by computing the middle point between higher and lower loss clusters:

$$\tau^{(t)} = \max\left\{ \zeta_i \mid v_i \in \mathcal{C}_{\mathrm{low}}^{(t)} \right\} + \frac{\min\left\{ \zeta_j \mid v_j \in \mathcal{C}_{\mathrm{high}}^{(t)} \right\} - \max\left\{ \zeta_i \mid v_i \in \mathcal{C}_{\mathrm{low}}^{(t)} \right\}}{2}. \tag{6}$$

To find the optimal epoch $t^*$ that best splits the target nodes, we first make an assumption:

**Assumption 1.** *There exists an optimal epoch during training at which the loss divergence between target nodes and clean nodes is maximized and is less pronounced or absent under standard Cross-Entropy loss. A theoretical justification of Assumption 1 is provided in Appendix C.2.*

**Theorem 1.** *Under the conditions of Assumption 1, there exists an optimal epoch $t^* \in [0, T_S]$, where $T_S$ is the number of early training epochs, such that the separation between target nodes and clean nodes is maximized.*

Given the existence of an optimal epoch, we compute the difference between the expectation of RCE loss value of the two clusters at each early epoch $t$. The epoch that maximizes this difference is selected as the optimal epoch $t^*$:

$$t^* = \arg\max_t \left( \mathbb{E}_{v_i \in \mathcal{C}_{\text{high}}^{(t)}} [\ell_i^{(t)}] - \mathbb{E}_{v_j \in \mathcal{C}_{\text{low}}^{(t)}} [\ell_j^{(t)}] \right). \tag{7}$$

We then treat nodes with $\zeta_i^{(t^*)} \leq \tau^{(t^*)}$ as candidate target nodes, while choosing the remaining nodes in training set as clean nodes:

$$\mathcal{V}_B^{S(t^*)} = \left\{ v_i \in \mathcal{V}_{y_t}^{(t^*)} \mid \zeta_i^{(t^*)} \leq \tau^{(t^*)} \right\}, \quad \mathcal{V}_C^{S(t^*)} = \mathcal{V}_T \setminus \mathcal{V}_B^{S(t^*)}. \tag{8}$$

After just a few epochs, the epoch-wise RCE loss distribution in Fig. 4 demonstrates a clear splitting, indicating that our proposed method can be effectively leveraged to identify target nodes. More results on other datasets are in Appendix G.

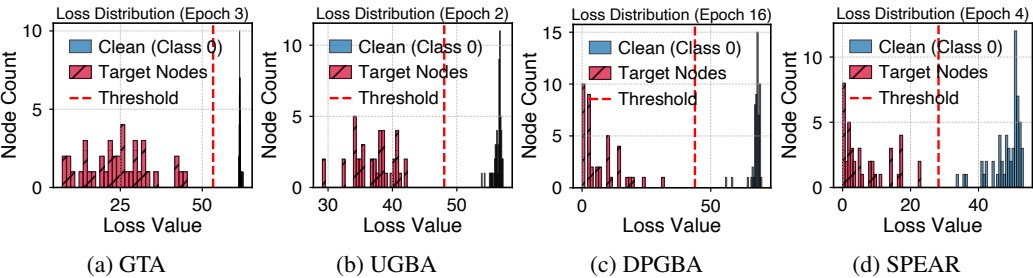

|          |          |          |          |
| :------: | :------: | :------: | :------: |
| (a) GTA  | (b) UGBA | (c) DPGBA | (d) SPEAR |

Figure 4: Epoch-wise loss distribution of nodes in target class 0 on Cora. Nodes in the left of the red dash line correspond to the candidate target nodes, while the right side are clean nodes with the same label as target nodes. The black color for clean node are actually blue because of severe overlapping.

## 4.2 Backdoor Recovery via Decoupling-Forgetting

Once the target node set $\mathcal{V}_B^{S(t^*)}$ is identified, the final step is to remove the backdoor effect by decoupling them from the malicious label $y_t$. We propose a unified Decoupling–Forgetting strategy that combines random label reassignment with gradient ascent.

**Random Label Reassignment.** Backdoor triggers establish a strong shortcut association between target nodes and malicious labels. To break this, we reassign a new random label except malicious label $\tilde{y}_i \in \mathcal{Y}_T' \setminus \{y_t\}$ to each target node at each training epoch. Unlike static reassignment, this dynamic relabeling acts as a diverse perturbation across epochs, preventing target nodes from overfitting to any single incorrect label.

**Gradient Ascent.** Although random relabeling prevents shortcut effect, it may cause the representation of the target nodes to converge towards the centroid of embedding space, which could occasionally align with certain malicious classes. To counteract this, we incorporate gradient ascent with respect to the malicious label $y_t$. This explicitly pushes the representation of target nodes away from the malicious decision boundary, resulting in a stronger and more directional forgetting signal.

**Unified Objective.** For identified target nodes $v_i \in \mathcal{V}_B^{S(t^*)}$, we employ a unified objective that combines *random relabeling* and *gradient ascent*, balanced by a trade-off parameter $\gamma$. In contrast, clean nodes $v_j \in \mathcal{V}_C^{S(t^*)}$ are trained normally to preserve model performance:

$$\min_\theta \mathcal{L}_\theta = \gamma \underbrace{\sum_{v_i \in \mathcal{V}_B^{S(t^*)}} \mathcal{L}\big(f_\theta(v_i), \tilde{y}_i\big)}_{\text{Random Relabeling}} + (1-\gamma) \underbrace{\sum_{v_i \in \mathcal{V}_B^{S(t^*)}} -\mathcal{L}\big(f_\theta(v_i), y_t\big)}_{\text{Gradient Ascent}} + \underbrace{\sum_{v_j \in \mathcal{V}_C^{S(t^*)}} \mathcal{L}\big(f_\theta(v_j), y_j\big)}_{\text{Normal Training}},$$

(9)

where $\tilde{y}_i \neq y_t$ is a random label exclude malicious label.

To further verify the superiority of our Decoupling-Forgetting strategy, we compare it with three intuitive method (Node Removal, Feature Reinitialization, and Restoring Original Labels) and one common competitve machine unlearning framework **SCRUB** [26]. For more details, please refer to Appendix E. The training algorithm and Time Complexity Analysis is in Appendix K and B.

## 5 Experiment

We conduct extensive experiments on multiple real-world graph datasets to evaluate the effectiveness of our method in defending against diverse graph backdoor attacks. Our evaluation is designed to systematically answer the following research questions: **Q1:** How effective is *LoSplit* in mitigating different types of graph backdoor attacks, especially SPEAR? **Q2:** How precise can *LoSplit* identify and split target nodes and clean nodes? **Q3:** How does each component of the *LoSplit* contribute to its overall defense performance? **Q4:** How do different number of early epochs, different early learning rate and different trade-off coefficient $\gamma$ impact the performance of *LoSplit*. **Q5:** How does *LoSplit* perform on clean graphs, i.e., when it is unknown whether the graph is perturbed? **Q6:** How does *LoSplit* perform under different attack configurations, such as varying the malicious labels and the number of injected triggers (attack budget)?

### 5.1 Experimental Settings

**Datasets.** We evaluate the effectiveness of *LoSplit* on six widely used node classification benchmark datasets. These datasets are Cora, Citeseer, Pubmed, the three classic citation network [27], the Physics collaboration network [28], the Flickr social network [29], and a large-scale academic graph OGB-arXiv [30]. These datasets cover citation networks and large-scale academic graphs with diverse structural and feature characteristics, allowing us to assess the robustness of our defense across different data scale. Details about these datasets are provided in Appendix D.1.

**Attack Methods.** To comprehensively evaluate the robustness of our approach, we consider four representative and diverse backdoor attack methods. For attacks based on structural perturbation, we include GTA [10], UGBA [11], and DPGBA [12], which manipulate the graph topology by inserting trigger subgraphs or adding edges connected to the target nodes. For feature perturbation attack, we include SPEAR [13], which introduces imperceptible changes to node features as triggers without altering the graph structure. For each attack, we follow the parameter setting specified in their original paper. Additional descriptions are provided in Appendix D.2.

**Defense Baselines.** We compare *LoSplit* against a comprehensive set of competitive baselines, including Detection-Deletion defenses (Prune [11] and OD [12]), robust training based defenses such as RIGBD [17] and one training-time defense in image ABL [18], the latter was originally proposed in the image domain and also using loss dynamics to identify target nodes. We also include two robust GNNs, i.e. RobustGCN [31] and GNNGuard [32]. Full details of these defense baselines are provided in Appendix D.3.

**Implementation Details.** Following previous work [17], we use a 2-layer GCN as our backbone model. We then split the perturbed graph into a training subgraph $\mathcal{G}_T$ and an unseen subgraph $\mathcal{G}_U$ using an 80:20 ratio. The attacker embeds triggers into selected target nodes in $\mathcal{G}_T$, and thus forms a trigger-embedded training graph $\mathcal{G}'_T$. Half of the nodes in $\mathcal{G}_U$ are selected as target nodes to compute the Attack Success Rate (ASR), while the remaining clean nodes are used to measure Clean Accuracy (CA). We also evaluate the Recall, Precision, and False Positive Rate (FPR) to demonstrate how precise we split target nodes, where Recall measures the proportion of correctly identified target nodes among all true target nodes, Precision measures the proportion of correctly identified target nodes among all identified candidates, and FPR indicates the proportion of clean nodes incorrectly

Table 1: Results of backdoor defense. Best results are in **bold**. Underlined results indicate that the highest Clean Accuracy (CA) does not coincide with the lowest ASR, and vice versa. This underscores that an effective defense must achieve both low ASR and high CA simultaneously.

| Attack | Defense | Cora | | CiteSeer | | PubMed | | Physics | | Flickr | | OGB-arXiv | |
|---|---|---|---|---|---|---|---|---|---|---|---|---|---|
| | | ASR(%)↓ | CA(%)↑ | ASR(%)↓ | CA(%)↑ | ASR(%)↓ | CA(%)↑ | ASR(%)↓ | CA(%)↑ | ASR(%)↓ | CA(%)↑ | ASR(%)↓ | CA(%)↑ |
| GTA | GCN | 98.52 | 82.96 | 99.40 | 73.80 | 97.62 | 84.53 | 100.00 | 96.23 | 100.00 | 42.39 | 94.70 | 63.12 |
| | RobustGCN | 100.00 | 81.85 | 99.70 | 73.49 | 97.87 | 85.19 | 100.00 | 94.98 | 99.89 | 40.44 | 99.83 | 60.16 |
| | GNNGuard | 38.38 | 75.19 | 12.31 | 62.95 | 21.35 | 81.33 | 80.94 | 96.35 | 0.24 | 43.75 | 0.88 | 63.42 |
| | Prune | 12.88 | 82.22 | 13.21 | 71.39 | 21.10 | 85.08 | 1.16 | 95.42 | 0.00 | 40.41 | 0.01 | 62.45 |
| | OD | 0.37 | 81.85 | **0.00** | 74.10 | 0.90 | 84.63 | **0.00** | 96.36 | 0.00 | 41.47 | **0.00** | 63.31 |
| | ABL | 4.80 | 78.52 | 1.50 | 73.19 | 1.77 | 83.71 | 100.00 | 96.25 | 0.00 | 40.80 | **0.00** | 63.92 |
| | RIGBD | 3.56 | 83.70 | **0.00** | 74.10 | 3.25 | 83.21 | 100.00 | 96.43 | 0.00 | 43.98 | **0.00** | 63.07 |
| | LoSplit | **0.00** | **84.81** | **0.00** | **75.60** | **0.06** | **85.29** | 0.56 | **96.43** | **0.00** | **44.19** | **0.00** | **65.74** |
| UGBA | GCN | 98.52 | 83.70 | 100.00 | 74.10 | 98.97 | 84.88 | 100.00 | 96.26 | 100.00 | 40.68 | 99.08 | 65.65 |
| | RobustGCN | 94.10 | 80.37 | 100.00 | 6.63 | 95.84 | **85.59** | 99.98 | 95.23 | 90.25 | 40.34 | 87.13 | 60.87 |
| | GNNGuard | 99.63 | 77.78 | 100.00 | 6.63 | 69.83 | 82.19 | 97.86 | 96.06 | 99.07 | 40.80 | 96.21 | 65.51 |
| | Prune | 98.52 | 78.52 | 96.70 | 72.89 | 88.29 | 85.08 | 95.73 | 95.16 | 90.23 | 40.45 | 93.99 | 64.46 |
| | OD | 12.92 | 83.70 | **0.00** | 75.30 | 83.98 | 84.88 | **0.00** | 96.20 | **0.00** | 40.25 | 10.13 | 65.32 |
| | ABL | 6.64 | 78.15 | **0.00** | 71.69 | 3.35 | 83.41 | 1.93 | 95.19 | **0.00** | 36.85 | 6.45 | 63.26 |
| | RIGBD | 7.11 | 83.70 | **0.00** | 73.49 | 2.54 | 82.65 | 0.56 | 96.38 | **0.00** | 40.58 | **0.00** | 66.06 |
| | LoSplit | **0.00** | **85.07** | **0.00** | **75.60** | **0.00** | 85.23 | 0.14 | **96.57** | **0.00** | **40.94** | **0.00** | **66.52** |
| DPGBA | GCN | 98.67 | 84.44 | 98.66 | 73.49 | 97.88 | 85.19 | 100.00 | 96.58 | 99.98 | 40.29 | 93.12 | 65.47 |
| | RobustGCN | 97.79 | 84.65 | 100.00 | **74.40** | 99.52 | 84.86 | 94.44 | 96.35 | 95.61 | 40.95 | 87.29 | 60.07 |
| | GNNGuard | 99.63 | 78.15 | 99.70 | 62.95 | 72.97 | 81.28 | 95.59 | 95.74 | 4.50 | 40.46 | 90.39 | 63.17 |
| | Prune | 22.88 | 79.63 | 11.41 | 72.89 | 40.92 | 84.53 | 1.61 | 96.23 | 0.00 | 40.62 | 0.12 | 62.76 |
| | OD | 96.31 | 81.85 | 97.90 | 74.10 | 84.89 | 81.13 | 94.52 | 96.25 | 98.56 | 40.59 | 94.21 | 65.06 |
| | ABL | 4.80 | 81.85 | **0.00** | 71.99 | 5.22 | 76.86 | 81.85 | 93.30 | 50.16 | 40.26 | 3.91 | 55.10 |
| | RIGBD | 2.22 | 84.07 | 0.30 | **74.40** | 4.92 | 84.37 | 0.98 | 96.27 | **0.00** | 40.78 | 11.83 | 63.43 |
| | LoSplit | **0.00** | **85.56** | **0.00** | **74.40** | 1.93 | **84.93** | **0.00** | **96.52** | **0.00** | **41.24** | **0.00** | **65.24** |
| SPEAR | GCN | 100.00 | 81.85 | 99.10 | 73.49 | 97.87 | 84.98 | 95.36 | 96.27 | 100.00 | 45.56 | 98.98 | 66.38 |
| | RobustGCN | 100.00 | 16.30 | 91.59 | 74.40 | 93.61 | **85.44** | 90.91 | 96.30 | 98.91 | 40.43 | 53.44 | 62.08 |
| | GNNGuard | 53.51 | 80.37 | 29.72 | 62.95 | 62.73 | 81.84 | 63.48 | 96.14 | 71.84 | 44.64 | 94.60 | **66.79** |
| | Prune | 100.00 | 84.07 | 100.00 | 72.29 | 98.83 | 85.19 | 96.78 | 96.15 | 100.00 | 40.52 | 99.83 | 65.77 |
| | OD | 100.00 | 80.00 | 100.00 | **76.50** | 94.48 | 85.29 | 53.92 | 96.22 | 41.59 | 41.48 | 66.31 | 66.31 |
| | ABL | 30.26 | 82.59 | **0.00** | 73.19 | 5.32 | 84.27 | 11.56 | 94.69 | 100.00 | 40.59 | 11.75 | 62.31 |
| | RIGBD | 97.78 | 83.70 | 90.27 | 72.29 | 88.98 | 84.68 | 88.03 | 96.35 | 100.00 | 44.24 | 97.09 | 66.72 |
| | LoSplit | **0.00** | **84.44** | **0.00** | 75.00 | **0.25** | 85.24 | **0.00** | **96.42** | **0.00** | **45.80** | 0.20 | 66.68 |

classified as target nodes. The trigger number $|\mathcal{V}_B|$ is set as 40, 40, 160, 160, 160 and 565 for Cora, Citeseer, PubMed, Physics, Flickr and OGB-arXiv, respectively.

## 5.2 Performance of Defense against Attacks (Q1)

To address **Q1**, we evaluate the effectiveness of defenses against multiple graph backdoor attacks. As summarized in Table 1, *LoSplit* consistently achieves state-of-the-art performance across all settings. Edge-dropping defenses such as Prune, OD and RIGBD are less effective against feature perturbation attacks (SPEAR). Though RIGBD performs well in its original paper, its performance degrades due to the sensitivity of its two hyperparameters, both of which requiring careful tuning for each dataset and attack. Random edge dropping may occasionally leave critical trigger edges, reducing ability in target nodes identification. Moreover, its heuristic threshold selection will also break down when two consecutive clean nodes are mixed with target nodes or when target candidates followed by clean nodes within the same class, leading to unreliable separation. ABL shows inferior performance compared to our approach due to the use of CE loss and fixed isolation ratio. These results highlight the superiority of *LoSplit* in defending against versatile graph backdoor attacks. Additional results using different GNN backbones (i.e. GAT [7] and GraphSage [33]) can be found in Appendix J.

## 5.3 Ability to Identify Target Nodes (Q2)

To address Q2, we assess the effectiveness of *LoSplit* in identifying target nodes by reporting Precision (Prec.), Recall (Rec.), and False Positive Rate (FPR) on Cora, Citeseer, and PubMed, as summarized in Table 2. Across all datasets, *LoSplit* consistently achieves near-perfect Precision and Recall while maintaining an extremely low FPR, indicating that it can precisely distinguish target nodes from clean ones with minimal misclassification. In contrast to existing defenses such as ABL and RIGBD, which often suffer from under-detection (i.e., low recall, failing to identify a portion of target nodes) or over-detection (i.e., high FPR, mistakenly treating clean nodes as targets), our design enables robust and adaptive detection under both structural and feature perturbation attacks. Additional experimental results on other datasets are provided in Appendix F.

Table 2: The Performance of Target Nodes Split on Cora, Citeseer, and PubMed (%). Best results are in **bold**. Underlined results indicate cases where these metrics do not align, i.e., achieving highest Precision or Recall does not correspond to the lowest FPR, highlighting that a perfect split must balance all three metrics simultaneously.

| Attack | Defense | Cora | | | Citeseer | | | PubMed | | |
|---|---|---|---|---|---|---|---|---|---|---|
| | | Prec. ↑ | Rec. ↑ | FPR ↓ | Prec. ↑ | Rec. ↑ | FPR ↓ | Prec. ↑ | Rec. ↑ | FPR ↓ |
| GTA | ABL | **100.00** | 85.00 | **0.00** | 88.10 | 92.50 | 0.75 | 24.39 | **100.00** | 3.05 |
| | RIGBD | 85.00 | 85.00 | 1.11 | 95.00 | 95.00 | 0.30 | 93.75 | 93.75 | 0.25 |
| | LoSplit | **100.00** | **100.00** | **0.00** | **100.00** | **97.50** | **0.00** | **98.77** | **100.00** | **0.05** |
| UGBA | ABL | **100.00** | 27.50 | **0.00** | **100.00** | 35.00 | **0.00** | **100.00** | 51.25 | **0.00** |
| | RIGBD | 72.50 | 72.50 | 2.03 | 77.50 | 77.50 | 1.35 | 92.41 | 92.41 | 0.30 |
| | LoSplit | **100.00** | **100.00** | **0.00** | **100.00** | **100.00** | **0.00** | **100.00** | **100.00** | **0.00** |
| DPGBA | ABL | **100.00** | 42.50 | **0.00** | **100.00** | 52.50 | **0.00** | 19.51 | 60.00 | 2.44 |
| | RIGBD | 81.08 | 75.00 | 1.29 | 97.06 | 82.50 | 0.15 | 92.06 | 72.50 | 0.25 |
| | LoSplit | **100.00** | **100.00** | 0.18 | **100.00** | **97.50** | **0.00** | **100.00** | **83.12** | **0.00** |
| SPEAR | ABL | **100.00** | 12.50 | **0.00** | 97.14 | 85.00 | 0.15 | 40.00 | 40.00 | 0.60 |
| | RIGBD | 83.33 | 12.50 | 0.18 | **93.75** | 37.50 | **0.10** | 80.77 | 51.22 | 0.13 |
| | LoSplit | **100.00** | **100.00** | **0.00** | 93.02 | **100.00** | 0.45 | **88.90** | **100.00** | **0.05** |

## 5.4 Ablation Study (Q3)

To assess **Q3**, we conduct an ablation study under the SPEAR attack on OGB-arxiv, evaluating four variants. *LoSplit\C* replaces RCE loss with standard CE loss. *LoSplit\S* uses a fixed isolation ratio and global threshold (always splitting nodes in half) just as those in image instead of dynamic split. *LoSplit\D* skip the Decoupling-Forgetting stage and only uses identified clean nodes to retrain the model. *LoSplit\R* applies the robust training strategy from RIGBD [17] to mitigate backdoor effect. As shown in Fig. 5, the removal of RCE significantly increases ASR, confirming its role in amplifying early loss dynamics. Static ratio and global threshold degrade adaptability, while only using clean nodes to retrain fails to break the shortcut. Compared with RIGBD, our fine-tuning method achieves better ASR reduction and preserves higher clean accuracy, demonstrating

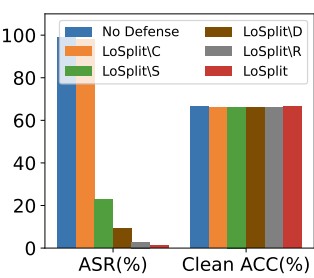

Figure 5: Ablation study.

that our Decoupling-Forgetting strategy is more effective than robust training strategy proposed in SOTA method RIGBD. These results underscore that all components synergistically improve the performance of *LoSplit*.

## 5.5 Hyperparameter Analysis (Q4)

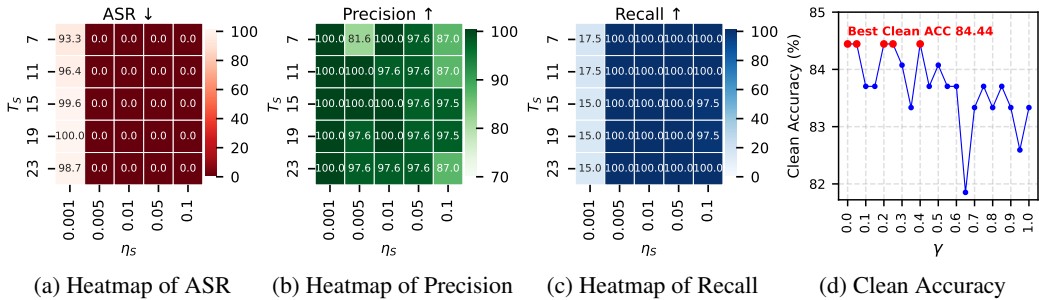

(a) Heatmap of ASR    (b) Heatmap of Precision    (c) Heatmap of Recall    (d) Clean Accuracy

Figure 6: Hyperparameter analysis of *LoSplit* on the Cora dataset under the SPEAR attack.

To address **Q4**, we analyze the impact of three key hyperparameters (the number of early-stage splitting epochs $T_S$, the early-stage learning rate $\eta_S$, and the trade-off coefficients $\gamma$ in the Decoupling–Forgetting stage.) in *LoSplit* on the Cora dataset under the SPEAR attack:

We observe that $T_S$ and $\eta_S$ mainly affect ASR, Precision, and Recall. We choose $T_S = \{3, 7, 11, 15, 19\}$ and $\eta_S = \{0.001, 0.005, 0.01, 0.05, 0.1\}$. Extremely low or high values of $T_S$ or $\eta_S$ either underfit or overfit the early loss dynamics, leading to degraded performance. As shown

in Fig. 6a-6c, optimal settings such as $T_S = \{7, 11, 15\}$ and $\eta_S = \{0.005, 0.01\}$ achieve impeccable split performance, yielding the lowest ASR (0.0%) and perfect precision and recall (100.0%).

Fixing $T_S = 11$ and $\eta_S = 0.005$, we then analyze the effect of $\gamma$, which control the balance between random relabeling and gradient ascent in the Decoupling-Forgetting, which primarily influence the clean accuracy (CA). We search $\gamma \in \{0, 0.05, 0.1, \ldots, 1.0\}$. As shown in Fig. 6d, the best result is achieved at $\gamma = \{0, 0.05, 0.2, 0.25, 0.4\}$, giving the best clean accuracy of 84.44%, achieving an improvement of more than 2.5% compared to when there is no dense (GCN).

Overall, *LoSplit* demonstrates strong robustness across a wide hyperparameter range, highlighting that our approach can achieve both high robustness (lowest ASR) and high utility (best Clean Accuracy). More results on other datasets are provided in Appendix M.

### 5.6 Performance on Clean Graphs (Q5)

To address **Q5**, we evaluate the behavior of our *LoSplit* defense when applied to clean graphs. Following the setup in Sec. 5.1, we remove all backdoor triggers from the training data and compare a standard 2-layer GCN (Without Defense) with our *LoSplit* defense.

The results are reported in Table 3. *LoSplit* achieves nearly the same accuracy as the GCN model across all datasets. This negligible gap indicates that our defense does not compromise model performance when no backdoor exists. At the same time, the False Positive Rate (FPR) is close to zero, meaning that nearly no clean nodes are mistakenly classified as target nodes.

The key reason for this behavior lies in our Early-Stage Dynamic Split strategy described in Sec. 4.1. In backdoored graphs, there will form a bimodal loss distribution within the target class. However, in clean graphs, such bimodal behavior vanishes so that no two cluster emerges. In this case, we manually set the threshold to an extremely small value (i.e., $1^{-10}$), effectively preventing any clean nodes from being misclassified.

Taken together, these results demonstrate that *LoSplit* is well-suited for practical scenarios where the contamination status of the graph is unknown.

Table 3: Performance of GCN and LoSplit on clean datasets in terms of clean accuracy (CA%) and False Positive Rate (FPR%).

|              | Cora  | Citeseer | PubMed | Physics | Flickr | OGB-arXiv |
|--------------|-------|----------|--------|---------|--------|-----------|
| GCN (CA)     | 83.70 | 74.70    | 85.18  | 96.02   | 45.33  | 66.12     |
| LoSplit (CA) | 83.33 | 74.39    | 85.03  | 95.87   | 45.11  | 65.98     |
| LoSplit (FPR)| 0.18  | 1.05     | 0.48   | 0.07    | 0.92   | 0.36      |

### 5.7 Performance under Different Malicious Labels and Attack Budget

For **Q6**, because of space limitation, the performance and analysis of *LoSplit* under different malicious labels and various attack budgets are presented in Appendix H and Appendix I. In these two scenarios, we encounter more complex situations where the separation between target nodes and clean nodes becomes less clear (i.e., severe overlapping, low precision). Even under such challenging conditions, *LoSplit* can still achieve strong defense performance owing to our Decoupling–Forgetting strategy and carefully-tuned hyperparameters.

## 6 Conclusion

In this paper, we propose *LoSplit*, the first training-time defense against graph backdoor attacks. By analyzing early-stage loss dynamics, we observe that target nodes and clean nodes within the same class form separable sub-clusters. Upon this insight, *LoSplit* splits target nodes and applies a Decoupling-Forgetting strategy to remove the backdoor influence. Experiments on various attacks and datasets show that *LoSplit* consistently improves both backdoor mitigation and model performance. While effective, its performance can be compromised when the separation between target nodes and clean nodes is less distinct and we have to tune the hyperparameters through trial-and-error which is often time-consuming and labor-intensive, limiting its robustness in highly diverse scenarios.

# 7 Acknowledgement

This work was supported by the National Natural Science Foundation of China (92370111, 62422210, 62302333, 62272340, 62276187), Hebei Natural Science Foundation (F2024202047).

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

# A  Details of Related Works

## A.1  Graph Backdoor Attacks

Graph backdoor attacks have emerged as a critical security threat in graph representation learning, particularly for GNNs. These attacks operate by injecting malicious triggers in the form of decently designed subgraphs or feature perturbations, into carefully selected training nodes. This creates a hidden shortcut between the perturbed target nodes and predetermined malicious labels during training. During inference, the infected model exhibits two distinct behaviors: it will consistently misclassify any input containing the trigger pattern to the attacker-specified label, while maintaining normal classification performance on unperturbed nodes. To improve the secrecy and effectiveness of such attacks, GTA [10] introduced a trainable trigger generator that produces sample-specific perturbations, substantially improving attack success rates. To improve this, UGBA [11] developed a more sophisticated target selection strategy that reduces the required number of target nodes while incorporating homophily constraints through enhacing the cosine similarity between triggers and target nodes, thereby improving both stealthiness and attack performance. To further boost stealthiness, DPGBA [12] addressed key limitations of existing methods by proposing an adversarial learning approach that generates in-distribution triggers and employs a novel loss function to boost attack success rates. While the attack above all pertub the topological structure of target nodes, SPEAR [13] represents a paradigm shift by exclusively perturbing node attributes while preserving the original graph topology. This approach significantly enhances stealthiness and presents unique detection challenges, causing traditional edge edge dropping based defense strategies ineffective. Therefore, our work focuses on defending against such highly stealthy attacks, particularly SPEAR where topological modifications are absent. Given the increasing sophistication of modern graph backdoor attacks and their potential security implications, developing robust defense mechanisms capable of detecting and mitigating such threat has become imperative.

## A.2  Graph Backdoor Defense

To mitigate the threat posed by graph backdoor attacks, a variety of defense strategies have been proposed, albeit still limited compared to the image domain. Existing methods primarily leverage structural inconsistencies introduced by triggers. For example, Prune [10] removes suspicious edges between nodes with low consine similarity, based on the observation that the trigger of attacks like GTA and UGBA often behave like out-of-distribution outliers, OD [12] adopts an unsupervised reconstruction-based approach using graph auto-encoders to identify and isolate outlier nodes with high reconstruction loss, which are likely to be injected triggers. To further generalize defense strategies across diverse attack settings, RIGBD [17] incorporates random edge dropping with robust training, aiming to train a benign model using poisoned training graph. Despite their effectiveness in structural perturbation scenario, these methods suffer from several limitations. In other words, these method are all edge dropping based, which will not be effective for SPEAR, and the heuristic method for splitting target nodes and clean nodes in RIGBD lacks robustness in particular situation. In contrast, our method exploits dynamic training signals, particularly the early-stage loss behavior, which is overlooked in existing graph domain defenses. Our training-time defense strategy provides a more fine-grained and adaptive mechanism for backdoor mitigation.

## A.3  Training-time Defense

Training-time defenses against backdoor attacks aim to detect and neutralize poisoned samples during the learning phase, thereby preventing the model from internalizing malicious correlations. Several recent approaches in vision domain have leveraged early training dynamics to isolate poisoned samples. For instance, ABL [18] observed that poisoned samples tend to exhibit lower training losses than benign ones. Based on this, they proposed a two-stage framework: initially identifying and isolating the samples with the lowest losses with fixed isolation ratio, followed by an gradient ascent unlearning phase aimed at mitigating the backdoor effect on these selected samples. DBD[19] and ASD [20], both using Symmetric Cross Entropy (SCE) Loss to separate poisoned samples, with ASD using a more precise meta-splitting method and then adopt hybrid training strategies, combining self-supervised or semi-supervised learning to mitigate the influence of poisoned data. PIPD [21] adopts a n-step progressive isolation strategy to iteratively isolate suspected poisoned data throughout training, improving model robustness without sacrificing clean accuracy. HARVEY [22] constructs a

strongly backdoored reference model by leveraging Reverse Cross-Entropy (RCE) loss to iteratively isolate poisoned samples. It begins with a naive training, progressively finetune the model to focus on backdoor patterns through learning and unlearning steps, and applies a meta-splitting strategy to refine the separation. The method introduces a paradigm shift by focusing on poisonous samples rather than benign ones and performs adaptive dataset splitting without a fixed splitting ratio. Finally, it retrains on the identified clean subset to obtain a backdoor-free model. While these methods have shown success in image classification tasks, they often overlook the unique challenges posed by graph-structured data such as homophily, message passing, and node interdependence where target nodes may propagate misleading signals to their neighbors. In the context of GNNs, training-time defense becomes more challenging due to the intertwined nature of the graph topology.

## B  Time Complexity Analysis

We analyze the time complexity of LoSplit by decomposing it into three key components, following the GCN complexity computation from [34] and [17]:

**(1) Early-stage Training.** The GNN is trained for $T_S < 20$ epochs on the trigger-embedded graph $\mathcal{G}_T$. For each layer, the time complexity is:

- Feature Transformation: $\mathcal{O}(NM^2)$ for $N$ nodes and $M$-dim features.
- Neighborhood Aggregation: $\mathcal{O}(|E|M)$ via sparse operations, where $|E|$ is the number of edges.

Over $L$-layer GCN and $T_S$ epochs, the total complexity for Early-Stage Training is:

$$\mathcal{O}\big(T_S \cdot L \cdot (NM^2 + |E|M)\big).$$

**(2) Loss Clustering.** For target-class nodes ($n_t$ nodes):

- Z-score standardization: $\mathcal{O}(n_t)$ per epoch.
- GMM clustering: $\mathcal{O}(n_t)$ (with $K = 2$, fixed parameter).

Over $T_S$ epochs, the whole complexity is $\mathcal{O}(T_S n_t)$.

**(3) Decoupling-Forgetting.** This stage shares the same formula as (1) but runs in $T >= 200$ epochs:

$$\mathcal{O}\big(T \cdot L \cdot (NM^2 + |E|M)\big).$$

**(4) Total Complexity.** Combining all stages, the total complexity is:

$$\mathcal{O}\big((T_S + T)L(NM^2 + |E|M) + T_S n_t\big).$$

We compare *LoSplit* with RIGBD as both of them follow a two-stage defense paradigm: identifying poisoned (target) nodes, followed by finetuning GNN model to eliminate the influence of backdoor. This shared structure places them in the same defense category. Under the SPEAR attack, *LoSplit* achieves substantially higher efficiency, as it only requires one full GCN training, one naive training on Early-stage Dynamic Split stage, and simple clustering, while RIGBD incurs the cost of two full trainings and repeated random edge perturbations. We run on an NVIDIA RTX 4090 GPU (24GB) with an Intel i7-13700K CPU, using Python 3.8.19 and PyTorch 1.12.1. According to Table 4, *LoSplit* outperforms RIGBD not only in defense effectiveness but also in computational efficiency. It requires only **42%**, **37%**, and **15%** the runtime of RIGBD on Cora, PubMed, and OGB-arXiv respectively, highlighting the scalability and practicality of our approach for large graphs.

Table 4: Running Time Comparison (seconds) with Complexity Analysis.

| Method | Complexity | Cora | PubMed | OGB-arXiv |
|---|---|---|---|---|
| RIGBD | $\mathcal{O}(L(K + 2T)(NM^2 + |E|M))$ | 4.36s | 18.69s | 917.12s |
| LoSplit | $\mathcal{O}(L(T_S+T)(NM^2 + |E|M) + T_S n_t)$ | 1.82s | 6.88s | 138.65s |

# C Detailed Proofs

## C.1 Proof of Why Target Nodes Converge faster than Clean nodes and Why RCE Loss Amplifies Such Behavior

To understand why the Reverse Cross-Entropy (RCE) loss amplifies early-stage loss dynamics better than the standard Cross-Entropy (CE) loss, we analyze and compare their gradient behavior.

**Cross-Entropy Loss (CE).** The CE loss is defined as:

$$\mathcal{L}_{\mathrm{CE}} = -\sum_{k=0}^{K-1} q(k|x) \cdot \log p(k|x), \tag{10}$$

where $q(k|x)$ is the one-hot ground-truth label distribution, and $p(k|x)$ is the predicted probability over $K$ classes.

**Reverse Cross-Entropy Loss (RCE).** The RCE loss is defined as:

$$\mathcal{L}_{\mathrm{RCE}} = -\sum_{k=0}^{K-1} p(k|x) \cdot \log q(k|x) \tag{11}$$

$$= -p(y|x) \cdot \log 1 + \sum_{k \neq y} p(k|x) \cdot C \tag{12}$$

$$= C \cdot (1 - p(y|x)), \tag{13}$$

where $y$ is the ground-truth label, and $C = -\log \varepsilon$ with $\varepsilon = 1^{-10}$, which prevents $\log 0$ when $q(k|x) = 0$.

We now compute the derivative of both loss functions with respect to the predicted probability $p(y_t|x)$ for the target class $y_t$:

$$\frac{\partial \mathcal{L}_{\mathrm{CE}}}{\partial p(y_t|x)} = -\frac{1}{p(y_t|x)}, \tag{14}$$

$$\frac{\partial \mathcal{L}_{\mathrm{RCE}}}{\partial p(y_t|x)} = -C. \tag{15}$$

Note that while CE loss exhibits a diminishing gradient as $p(y_t|x)$ increases, the RCE loss maintains a constant gradient magnitude due to the constant coefficient $C$. This property ensures stronger gradient feedback for confident predictions in early training.

Assuming a 1-layer GCN, let $z = \sigma(H_u)$ be the pre-softmax logits where $\sigma$ is the ReLU function. The predicted probability $p(y_t|x) = \mathrm{softmax}(z)$. Then the embedding $H_u$ for a target node $u$ is:

$$H_u = \sum_{v \in \mathcal{N}(u)} \frac{1}{\sqrt{d_v d_u}} X_v W_B + \frac{1}{d_u} X_u W_B + \frac{1}{\sqrt{d_u d_\delta}} \delta W_B, \tag{16}$$

or in the case of feature perturbation:

$$H_u = \sum_{v \in \mathcal{N}(u)} \frac{1}{\sqrt{d_v d_u}} X_v W_C + \frac{1}{d_u} (X_u + \delta) W_C, \tag{17}$$

where $\delta$ denotes the attributes of embedded trigger, and $W_B, W_C$ are trainable weights for target nodes and clean nodes, respectively.

For a clean node $v$, the embedding is:

$$H_v = \sum_{u \in \mathcal{N}(v)} \frac{1}{\sqrt{d_u d_v}} X_u W_C + \frac{1}{d_v} X_v W_C. \tag{18}$$

We now compute gradients using the chain rule. First, the derivative of $p(y_t|x)$ with respect to logits $z_j$ is:

$$\frac{\partial p(y_t|x)}{\partial z_j} = \begin{cases} p(y_t|x)(1 - p(y_t|x)) & \text{if } j = y_t, \\ -p(y_t|x) \cdot p(j|x) & \text{if } j \neq y_t. \end{cases} \tag{19}$$

Next, we have:

$$\frac{\partial z}{\partial H_u} = 1, \tag{20}$$

and for backpropagation through the linear layer:

$$\frac{\partial H_u}{\partial W_B} = \sum_{v \in \mathcal{N}(u)} \frac{1}{\sqrt{d_v d_u}} X_v + \frac{1}{d_u} X_u + \frac{1}{\sqrt{d_u d_\delta}} \delta, \tag{21}$$

or for feature perturbation:

$$\frac{\partial H_u}{\partial W_B} = \sum_{v \in \mathcal{N}(u)} \frac{1}{\sqrt{d_v d_u}} X_v + \frac{1}{d_u} (X_u + \delta). \tag{22}$$

Similarly, for a clean node $v$, we have:

$$\frac{\partial H_v}{\partial W_C} = \sum_{u \in \mathcal{N}(v)} \frac{1}{\sqrt{d_u d_v}} X_u + \frac{1}{d_v} X_v. \tag{23}$$

Let us define:

$$S_B = \frac{\partial H_u}{\partial W_B}, \quad S_C = \frac{\partial H_v}{\partial W_C}.$$

Then the gradient of the loss of target nodes with respect to weights are:

$$\frac{\partial \mathcal{L}_{\text{CE}}}{\partial W_B} = (p(y_t|x) - 1) \cdot S_B, \tag{24}$$

$$\frac{\partial \mathcal{L}_{\text{RCE}}}{\partial W_B} = C \cdot p(y_t|x)(p(y_t|x) - 1) \cdot S_B. \tag{25}$$

The gradient of clean nodes:

$$\frac{\partial \mathcal{L}_{\text{CE}}}{\partial W_C} = (p(y_c|x) - 1) \cdot S_C, \tag{26}$$

$$\frac{\partial \mathcal{L}_{\text{RCE}}}{\partial W_C} = C \cdot p(y_c|x)(p(y_c|x) - 1) \cdot S_C. \tag{27}$$

**Why Target Nodes Converge Faster During Early Training.** Target nodes are connected to trigger-embedded nodes or features with deliberately crafted attributes $\delta$, which exhibit an exceptionally strong correlation with the malicious label $y_t$. As a result, $S_B$ becomes heavily dominated by $\delta$, biasing $p(y_t|x)$ upward during the early stages of training. From the gradient perspective, when $p(y_t|x) < 0.5$, the slope of the loss curve with respect to $p(y_t|x)$ is much steeper for target nodes due to the strong influence of $\delta$, resulting in larger gradient magnitudes and faster convergence. In contrast, clean nodes with $S_C$ lack such strong and consistent bias, yielding smaller gradients and slower updates (i.e., $|S_C| \ll |S_B|$).

**Why RCE Loss Amplifies This Behavior.** As illustrated in Fig. 7, the gradient magnitude of the RCE loss is substantially larger than that of the CE loss when $p(y_t|x) < 0.5$, i.e., during the early stages of training. While CE loss decreases its gradient linearly as the model becomes more confident, RCE imposes a much steeper penalty for low-confidence predictions. This sharp gradient slope drives faster updates for target nodes that are initially misclassified, further accelerating their convergence toward the malicious label. Consequently, RCE amplifies the early-stage dynamics observed in target nodes, making them more distinguishable from clean nodes during early training.

## C.2 Proof of Assumption 1

We provide a theoretical justification for the existence of an optimal epoch $t^*$ during early training with Reverse Cross-Entropy (RCE) loss, at which the loss divergence between target nodes and clean nodes is maximized. This underpins our assumption that early-stage loss dynamics are informative for splitting target nodes and clean nodes.

**Gradient Behavior Under CE loss and RCE loss.** The gradient of the CE loss with respect to model weights $W_B$ for a target node with respect to the prediction $p(y_t|x)$ (Eq. 24) is largest in magnitude when $p(y_t|x) \to 0$, and vanishes as $p(y_t|x) \to 1$. Thus, CE loss provides diminishing gradient signals for target nodes that converge early. By contrast, the RCE loss gradient in Eq. 25 is a concave quadratic function with respect to $p(y_t|x)$, which reaches its maximum magnitude at $p(y_t|x) = 0.5$, and vanishes at both $p(y_t|x) \to 0$ and $p(y_t|x) \to 1$ as illustrated in Fig. 7.

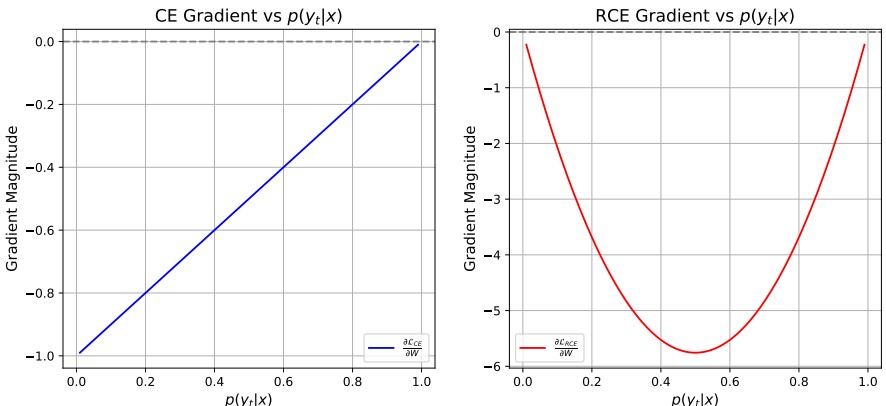

Figure 7: Gradient magnitude (w.r.t. prediction confidence $p(y_t|x)$) for Cross-Entropy (CE) and Reverse Cross-Entropy (RCE). Unlike CE, RCE induces a peak gradient at $p = 0.5$, which amplifies early-stage separation between fast-converging target nodes and slower clean nodes.

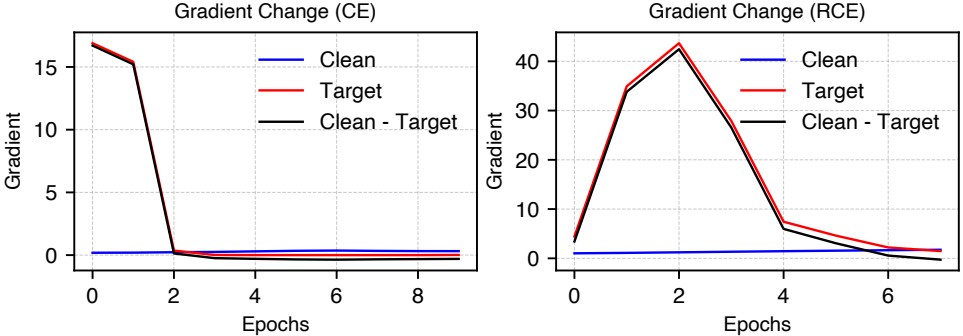

Figure 8: Gradient dynamics over training epochs under Cross-Entropy (left) and Reverse Cross-Entropy (right). Under CE, both clean and target node gradients quickly decay and overlap. Under RCE, target node gradients exhibit a transient peak in early epochs while clean gradients remain low, producing a clear early-stage separation.

**Why an Optimal Epoch $t^*$ Exists.** During early training, both clean nodes and target nodes start with low prediction confidence. However, due to the connection to backdoor triggers, target nodes quickly converge toward the target class $y_t$, surpassing the $p(y_t|x) = 0.5$ region and entering the gradient-suppressed regime of RCE loss.

In contrast, clean nodes lack such shortcut connections, progress more slowly and remain within the high-gradient region centered around $p(y_t|x) = 0.5$. As a result, the RCE loss of clean nodes temporarily increases (or decreases slowly), while the loss of target nodes decreases sharply.

These theoretical insights are empirically supported in Fig. 8, which compares the per-epoch gradient magnitudes of clean nodes and target nodes under CE loss and RCE loss. CE loss quickly flattens the gradient difference, while RCE loss maintains a transient peak in target gradients, creating a temporal separation window between the two node types.

This mismatch in the speed of convergence causes the loss gap between the two groups widen during early epochs and then shrink as both converges just as the black line in Fig. 8. Theoretically, we have:

$$\Delta(t) = \mathbb{E}_{v \in \mathcal{V}_C}[\mathcal{L}_{\text{RCE}}^{(t)}(v)] - \mathbb{E}_{u \in \mathcal{V}_B}[\mathcal{L}_{\text{RCE}}^{(t)}(u)],$$

which denotes the expected RCE loss difference at epoch $t$ between clean nodes and target nodes. Since this equation increases and then decreases during training, thus it will reach a maximum value at some intermediate epoch $t^*$, where the two groups are maximally separated in the loss space.

The non-monotonic gradient behavior of RCE loss induces a temporal window during which the loss values of target nodes and clean nodes diverge maximally, which guarantees the existence of an optimal early epoch $t^*$ that maximizes the separation between target nodes and clean nodes. Consequently, Our Assumption 1 is theoretically justified and supports the design of our Early-Stage Dynamic Split mechanism.

## D    Details of Experimental Settings

### D.1    Datasets Details

**Cora, Citeseer, and Pubmed.** These are three widely used citation network benchmarks in graph learning. In each dataset, nodes represent documents, and edges denote citation relationships. Cora contains 2,708 nodes and 5,429 edges, with each node described by a 1,433-dimensional bag-of-words feature vector across 7 classes. CiteSeer includes 3,327 nodes, 4,552 edges, and 3,703-dimensional features across 3 categories. PubMed consists of 19,717 nodes and 44,338 edges, with 500-dimensional TF/IDF-like features and 3 classes.

**Physics.** Coauthor Physics is a co-authorship graph based on the Microsoft Academic Graph from the KDD Cup 2016 challenge [28]. Nodes represent authors, connected by an edge if they co-authored a paper. Node features represent paper keywords for each author's papers, and class labels indicate the most active fields of study for each author. The dataset contains 34,493 nodes and 247,962 edges, with 8,415-dimensional features across 5 classes.

**Flickr.** Flickr [29] is a social network graph where each node represents an image uploaded to Flickr. Edges are established between images that share common properties such as geographic location, gallery, or user comments. Node features are represented by a 500-dimensional bag-of-words model from the NUS-wide dataset. We examined the 81 tags assigned to each image and manually consolidated them into 7 distinct classes, with each image belonging to one category. The dataset contains 89,250 nodes and 899,756 edges.

**OGB-arXiv.** OGB-arXiv [30] is provided by the Open Graph Benchmark, OGB-arXiv represents a large-scale citation network from the arXiv paper corpus. It includes 169,343 nodes and 1,166,243 edges. Each node is described by a 128-dimensional feature derived from title and abstract content, and classified into one of 40 scientific fields. It uses a temporal split for train/validation/test sets to better simulate real-world applications.

The details of the six datasets are provided in Table 5.

Table 5: Statistics of datasets used in our experiments.

| Dataset | Nodes | Edges | Features | Classes |
|---|---|---|---|---|
| Cora | 2,708 | 5,429 | 1,433 | 7 |
| CiteSeer | 3,327 | 4,552 | 3,703 | 3 |
| PubMed | 19,717 | 44,338 | 500 | 3 |
| Physics | 34,493 | 247,962 | 8,415 | 5 |
| Flickr | 89,250 | 899,756 | 500 | 7 |
| OGB-arXiv | 169,343 | 1,166,243 | 128 | 40 |

## D.2 Attack Method

We briefly describe the four backdoor attack methods evaluated in this paper.

**GTA**: GTA [10] is the first method to leverage a learnable trigger generator that produces sample-specific subgraph triggers for each target node. The generator is optimized to maximize the attack success rate (ASR), but it lacks constraints on stealthiness, which may lead to detectable patterns in the poisoned graphs.

**UGBA**: UGBA [11] enhances GTA by selecting a set of diverse and representative nodes for poisoning, improving attack efficiency. It also incorporates a homophily constraint that forces the generated trigger features to align with those of the target node's neighbors, improving stealth and making the backdoor more difficult to detect.

**DPGBA**: DPGBA [12] proposes an adversarial training framework to generate in-distribution triggers. A novel loss function ensures that the generated triggers remain close to the data distribution while still being effective. This design helps the attack remain stealthy and improves the overall ASR compared to prior methods.

**SPEAR**: SPEAR [13] targets the feature space instead of the graph structure by learning to perturb a small number of node features in a subtle but malicious way. It selects stealthy features and vulnerable nodes to inject the trigger. SPEAR is highly effective and particularly challenging to defend against, as it leaves the graph structure unchanged and manipulates only a few features.

## D.3 Defense Baseline

We evaluate the effectiveness of *LoSplit* against a range of representative defense baselines, which can be grouped into three categories:

**Detection-Deletion Defenses.** These methods aim to remove or weaken the influence of trigger structures in the graph:

- **Prune** [11]: Prune observes that trigger patterns often violate the homophily property commonly found in real-world graphs. It removes edges between nodes with low similarity, thereby disrupting trigger connections. Prune can be applied during both training and inference.

- **OD** [12]: For Outlier Detection (OD) method, we use DOMINANT [35] where a graph auto-encoder is used to reconstruct node features and filter out nodes with high reconstruction error. This helps eliminate anomalous triggers while keeping the clean ones. It is only applied during training.

**Robust Training Defense.** These defenses aim to improve the integrity of the model by first identify and isolate target nodes and then retrain the backdoored model:

- **RIGBD** [17]: RIGBD identifies candidate trigger nodes through random edge perturbations and performs adversarial training to suppress their influence. It enhances model robustness against structure-based backdoor attacks.

**Training-time Defense.**

- **ABL** [18]: We adapt the Anti-Backdoor Learning (ABL) framework, originally designed for DNNs, to GCN. ABL leverages two intrinsic weaknesses of backdoor attacks: (1) poisoned samples are typically learned much faster than clean ones, and stronger attacks further accelerate this convergence; (2) the backdoor objective is tightly associated with a specific target class. Based on these observations, ABL employs a Local Gradient Ascent (LGA) strategy to constrain the loss of each sample around a specific threshold. In parallel, a Global Gradient Ascent (GGA) objective is introduced to unlearn the backdoor associations by increasing the loss of poisoned sample while preserving performance on clean nodes.

**Robust GNNs.** These methods are originally designed for general adversarial attacks but can be adapted to backdoor scenarios:

- **GNNGuard** [32]: GNNGuard uses node feature similarity to adaptively reweight edges during training, effectively suppressing adversarial perturbations and enhancing model resilience.
- **RobustGCN** [31]: RobustGCN models node embeddings as Gaussian distributions and introduces variance-based attention to reduce the influence of anomalous neighbors, improving robustness against structure attacks.

# E   Comparison with Alternative Backdoor Recovery Strategies

To further validate the superiority of our proposed Decoupliung-Forgetting strategy that combines Random Label Reassignment and Gradient Ascent, we compare it with three intuitive methods, i.e. **Node Removal**, **Feature Reinitialization**, and **Restoring Original Label**, as well as one competitive machine unlearning framework **SCRUB** [26].

Table 6: Comparison of different backdoor recovery strategies under four representative attacks on PubMed dataset. Lower attack success rate (ASR, %) and higher clean accuracy (CA, %) indicate better performance.

| Strategy | GTA | | UGBA | | DPGBA | | SPEAR | |
|---|---|---|---|---|---|---|---|---|
| | ASR↓ | CA↑ | ASR↓ | CA↑ | ASR↓ | CA↑ | ASR↓ | CA↑ |
| GCN (No Defense) | 97.81 | 84.42 | 95.69 | 83.16 | 98.78 | 84.98 | 92.90 | **85.13** |
| Node Removal | 0.13 | 84.98 | 0.00 | 85.08 | 95.30 | **85.39** | 0.33 | 84.73 |
| Feature Reinitialization | 100.00 | 80.92 | 0.00 | 84.07 | 98.39 | 84.78 | 100.00 | 81.78 |
| Restore Original Label | **0.00** | 84.37 | 0.00 | 81.28 | **1.15** | 84.93 | 0.08 | 84.24 |
| SCRUB [26] | **0.00** | 84.58 | 0.00 | 82.90 | 97.75 | 84.63 | 0.00 | 84.47 |
| **Decoupling–Forgetting** | 0.06 | **85.19** | **0.00** | **85.33** | 1.92 | 84.93 | **0.00** | 85.13 |

From the results in Table 6, *LoSplit* consistently outperforms all four alternative backdoor recovery strategies. We summarize the main insights as follows:

**(1) Node Removal.** This strategy completely discards the identified target nodes, which indeed eliminates the backdoor effect but simultaneously disrupts the structural integrity of the graph. Such removal damages neighborhood information and leads to performance degradation especially in DPGBA attack ((95.30% ASR)) where triggers mimic benign neighborhood patterns, simple node deletion fails to fully suppress the trigger influence.

**(2) Feature Reinitialization.** This method assumes that the backdoor mainly resides in the feature space and thus resets the features of target nodes. However, resetting node features fails to break the association between target nodes and the malicious label, which explains why this strategy remains ineffective or even worse against GTA, DPGBA, and SPEAR attacks.

**(3) Restoring Original Label** appears to be an intuitive solution by relabeling the target candidates to their ground-truth classes. While effective in vision domains where poisoned samples (e.g., images with visible triggers) can be manually corrected at low cost, this approach becomes impractical in graphs. A node's label depends not only on its attributes but also on its local topology and neighborhood context, making it difficult to determine the true label without significant human effort. Consequently, label restoration is prohibitively expensive and rarely scalable for large graphs.

**(4) Mainstream Unlearning Framework.** Another competitive alternative is to adapt general machine unlearning frameworks such as SCRUB [26], which minimize both the cross-entropy and KL-divergence losses on the retain set (clean nodes), while maximizing the KL-divergence on the forget set (target nodes):

$$\min_{\theta} \mathcal{L}_{\theta}^{\text{SCRUB}} = \underbrace{\lambda_u \sum_{v_j \in \mathcal{V}_C^{S(t^*)}} \mathcal{L}\big(f_\theta(v_j), y_j\big) + \lambda_d \sum_{v_j \in \mathcal{V}_C^{S(t^*)}} \text{KL}(v_j)}_{\text{Retain Set (Clean Nodes)}} - \underbrace{\sum_{v_i \in \mathcal{V}_B^{S(t^*)}} \text{KL}(v_i)}_{\text{Forget Set (Target Nodes)}}, \quad (28)$$

where $\mathcal{V}_C^{S(t^*)}$ and $\mathcal{V}_B^{S(t^*)}$ denote the clean nodes(retain) and target nodes (forget) identified at epoch $t^*$, respectively. $\mathcal{L}$ is the cross-entropy loss used for label restoration, and $\text{KL}\big(p_{f_{\theta^T}}(v) \,\|\, p_{f_{\theta^S}}(v)\big)$ is

the Kullback–Leibler divergence between the teacher model $f_{\theta^{\text{old}}}$ and the student model $f_\theta$, defined as

$$\mathrm{KL}\big(p_{f_{\theta^T}}(v) \,\|\, p_{f_{\theta^S}}(v)\big) = \sum_k p_{f_{\theta^T}}^{(k)}(v) \log \frac{p_{f_{\theta^T}}^{(k)}(v)}{p_{f_{\theta^S}}^{(k)}(v)},$$

where $p_{f_{\theta^T}}(v)$ and $p_{f_{\theta^S}}(v)$ represent the teacher and student predictive distributions, respectively. This teacher–student design encourages the student to retain the teacher's knowledge on the clean (retain) set while deliberately forgetting it on the target (forget) set.

However, to maintain clean accuracy, SCRUB requires substantially more training epochs on the retain set during alternating optimization process, leading to significant computational cost. Even under its best settings, the overall robustness of SCRUB remains inferior to our Decoupling–Forgetting strategy.

Overall, our Decoupling–Forgetting approach achieves consistently lower ASR and competitive CA across all attacks while remaining lightweight and graph-specific, demonstrating a more effective and efficient mechanism for backdoor recovery in GNNs.

# F    Additional Results of The Ability to Split Target Nodes

In this section, we continue to evaluate the ablility of *LoSplit* in splitting target nodes on larger graphs. We observe consistent trends with those on Cora, Citeseer, and Pubmed (Sec. 5.3) in Table 7. First, *LoSplit* achieves near-perfect Precision and Recall across most attack settings, with FPR close to zero. For example, under UGBA attack, LoSplit reaches 100% Precision and Recall with 0% FPR across all three datasets, clearly outperforming ABL and RIGBD. In contrast, ABL often suffers from under-detection (e.g., Physics, where Recall drops below 7%) or over-detection (e.g., 97.5% in Recall but only 21.67% in Precision for Flickr dataset). RIGBD shows more balanced detection, but still fails in cases with complex perturbations (e.g., OGB-arXiv under DPGBA, only $4.42\%$ in Recall). Second, *LoSplit* maintains robustness across diverse attacks. Even under strong feature perturbation attacks like SPEAR, *LoSplit* achieves high detection quality. For all datasets, Precision and Recall all achieves over $90\%$ and FPR under $1\%$ This demonstrates that our method can effectively adapt to both structural and feature perturbations without significant degradation. Overall, these results confirm that the proposed Early-stage Dynamic Split design generalizes well beyond small citation graphs. *LoSplit* consistently outperforms ABL and RIGBD, achieving both high Precision and Recall while minimizing FPR, even on larger and challenging datasets.

Table 7: The Performance of Target Nodes Split (%) on Physics, Flickr, and OGB-arXiv. Best results are highlighted in **bold**. Underlined results indicate cases where the metrics are not consistent, meaning that achieving the highest Precision or Recall does not necessarily coincide with the lowest FPR. This suggests that an ideal split should maintain a balanced trade-off among all three metrics.

| Attack | Defense | Physics | | | Flickr | | | OGB-arXiv | | |
|---|---|---|---|---|---|---|---|---|---|---|
| | | Prec. ↑ | Rec. ↑ | FPR ↓ | Prec. ↑ | Rec. ↑ | FPR ↓ | Prec. ↑ | Rec. ↑ | FPR ↓ |
| | ABL | 12.50 | 10.00 | 0.92 | 21.67 | 97.50 | 0.78 | 8.14 | 70.00 | 0.92 |
| GTA | RIGBD | 87.14 | 75.00 | 0.18 | 93.13 | 93.13 | 0.61 | 97.17 | 97.17 | 0.05 |
| | LoSplit | **96.97** | **100.00** | **0.07** | **96.36** | **99.37** | **0.03** | **99.65** | **99.82** | **0.01** |
| | ABL | 8.33 | 6.25 | 0.95 | 21.67 | 97.50 | 0.78 | 7.72 | 62.50 | 0.93 |
| UGBA | RIGBD | 92.41 | 80.00 | 0.14 | 99.37 | 98.12 | 0.06 | 98.76 | 98.76 | 0.02 |
| | LoSplit | **100.00** | **100.00** | **0.00** | **100.00** | **100.00** | **0.00** | **100.00** | **100.00** | **0.00** |
| | ABL | 10.71 | 7.50 | 0.88 | 0.00 | 0.00 | 1.01 | 9.26 | 85.00 | 0.53 |
| DPGBA | RIGBD | 85.29 | 71.25 | 0.25 | 100.00 | 83.12 | 0.00 | 86.21 | 4.42 | 0.01 |
| | LoSplit | **96.97** | **100.00** | **0.07** | **100.00** | **88.12** | **0.00** | **100.00** | **97.50** | **0.00** |
| | ABL | 11.76 | 9.38 | 0.90 | 0.00 | 0.00 | 1.01 | 96.80 | 58.94 | 0.03 |
| SPEAR | RIGBD | 78.57 | 72.50 | **0.33** | 0.89 | **100.00** | **0.00** | 0.00 | 0.00 | **0.00** |
| | LoSplit | **92.12** | **90.50** | 0.56 | **100.00** | **100.00** | **0.00** | **99.45** | **96.81** | 0.01 |

# G    Additional Results of Split using Early-Loss Dynamics For Different Datasets

In this section, we present a further analysis of the target nodes split via early loss dynamics under other five datasets. As shown in Fig. 9–13, we observe that across all attack variants, target nodes

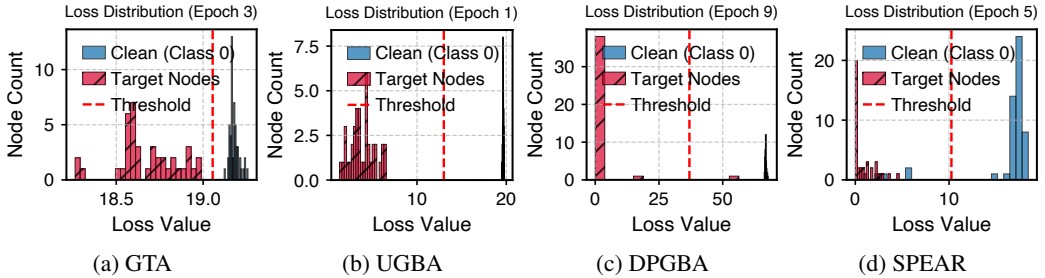

Figure 9: Distribution of the Early-Stage Dynamic Split in Target Class on **Citeseer**.

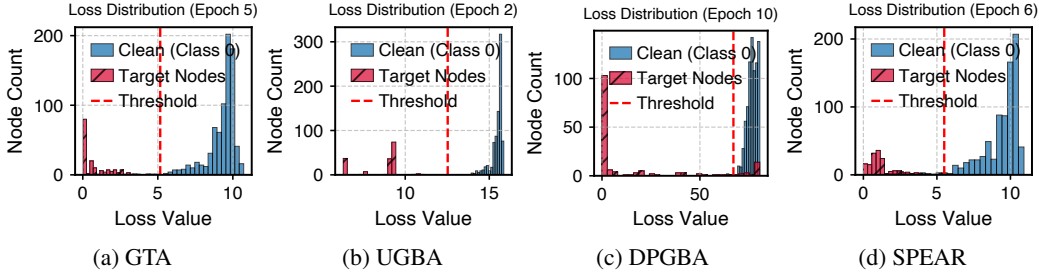

Figure 10: Distribution of the Early-Stage Dynamic Split in Target Class on **PubMed**.

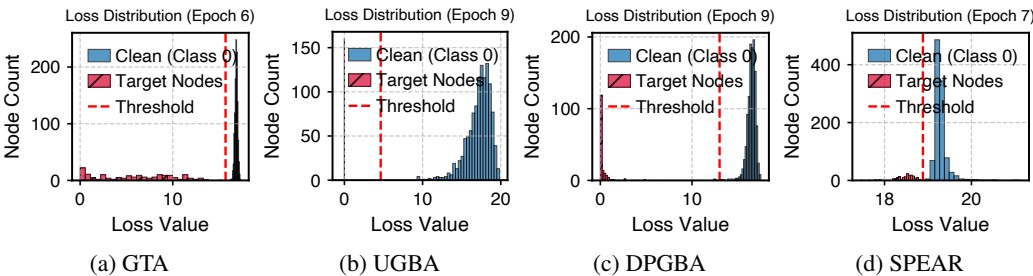

Figure 11: Distribution of the Early-Stage Dynamic Split in Target Class on **Physics**.

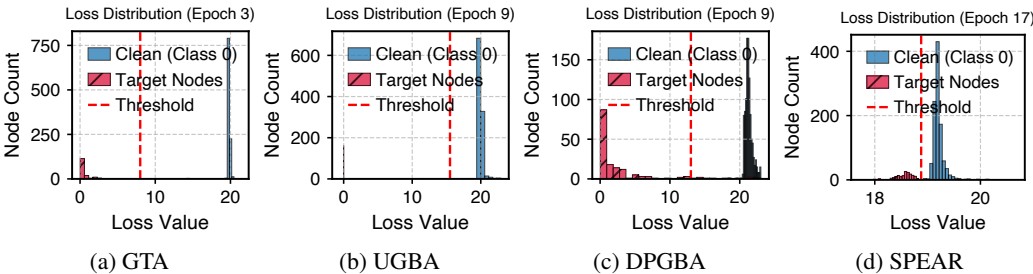

Figure 12: Distribution of the Early-Stage Dynamic Split in Target Class on **Flicker**.

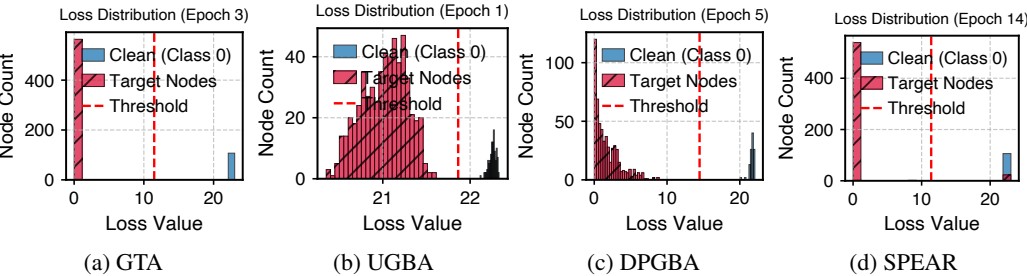

Figure 13: Distribution of the Early-Stage Dynamic Split in Target Class on **OGB-arXiv**.

tend to exhibit distinct early-stage loss distributions compared to clean nodes in the same class. This separation is particularly clear in GTA and UGBA, where the backdoor introduces a strong shortcut, leading to rapidly decreasing or highly concentrated loss values for target nodes. Even in more stealthy attacks such as DPGBA and SPEAR, which aim to minimize detectable differences in graph structure or features, but the early training dynamics still provides reliable signals for separating target nodes through careful tuning.

## H  Additional Results of Split using Early-Loss Dynamics Under Different Malicious Labels

We further assess the robustness of *LoSplit* by varying the attacker-specified target class $y_t$ categories. Since different classes may exhibit different learning dynamics—particularly those that are easier to learn, thus we aim to investigate whether *LoSplit* remains effective when the backdoor is injected into such easily learnable classes. This scenario is especially challenging because the clean nodes belonging to the target class may converge faster during training, potentially obscuring the divergence between target and clean nodes. Experimental results presented in Fig. 14 and Fig. 15, demonstrate that *LoSplit* consistently achieves clear separation, highlighting its robustness regardless of different malicious label.

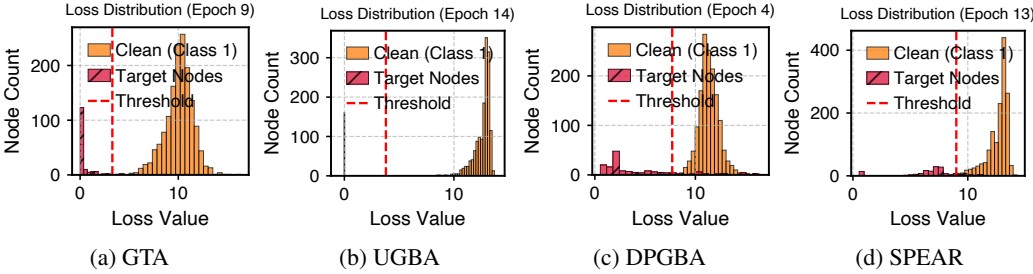

Figure 14: Results of Early-Loss Dynamics Split Under target label 1.

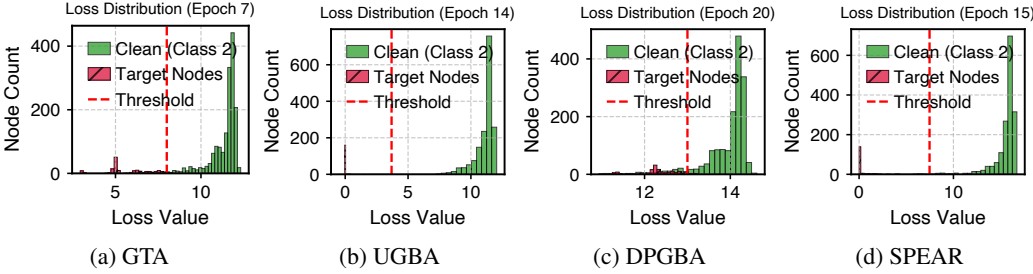

Figure 15: Results of Early-Loss Dynamics Split Under target label 2.

## I  Results of Defense Performance and Separation Ability using Early-Loss Dynamics under Different Trigger Numbers

To further evaluate the robustness of *LoSplit* against different attack budgets, we vary the number of backdoor triggers and examine its performance on both Cora and PubMed dataset under the SPEAR attack. Specifically, we adopt trigger numbers $\{10, 20, 40, 100, 160\}$ and $\{40, 80, 160, 200, 320\}$ for Cora and PubMed, respectively. The hyperparameters in Early-Stage Dynamic Split stage are set to $T_S = 10, \eta_S = 0.008$ for Cora, and for PubMed are carefully tuned under different trigger number setting: $T_S = 14, \eta_S = 0.008, \gamma = 0.9$ (40 triggers), $T_S = 20, \eta_S = 0.001, \gamma = 0.15$ (80 triggers), $T_S = 18, \eta_S = 0.001, \gamma = 0.35$ (160 triggers), $T_S = 18, \eta_S = 0.001, \gamma = 0.5$ (200 triggers), and $T_S = 5, \eta_S = 0.005, \gamma = 0.75$ (320 triggers).

As summarized in Table 8, *LoSplit* consistently achieves perfect ASR (0%) on the Cora dataset across all trigger numbers, demonstrating stable and strong defensive capability even as the backdoor budget

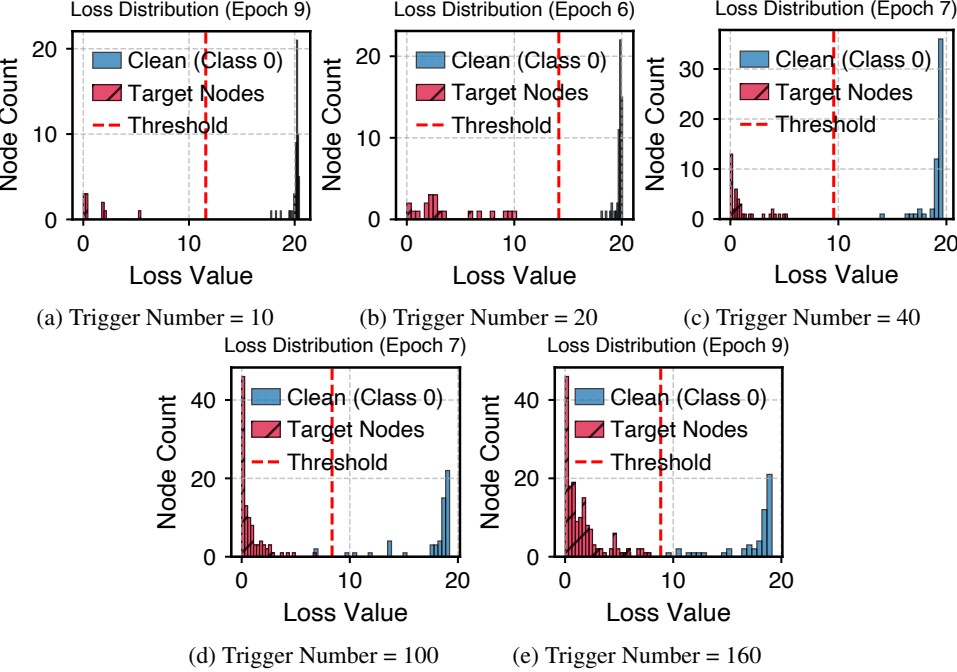

(a) Trigger Number = 10     (b) Trigger Number = 20     (c) Trigger Number = 40

(d) Trigger Number = 100     (e) Trigger Number = 160

Figure 16: Split on Cora under different attack budget.

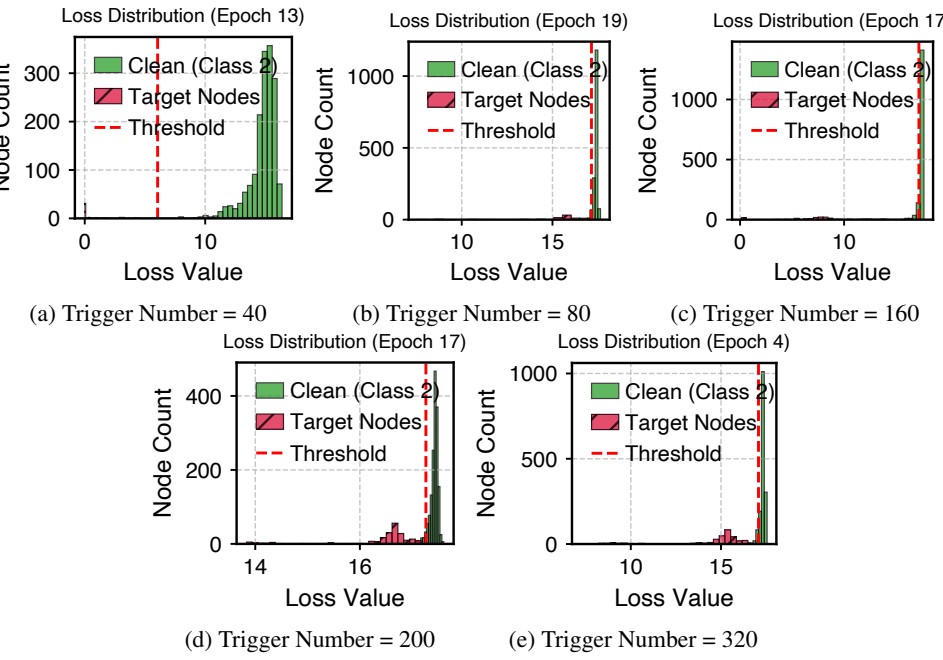

(a) Trigger Number = 40     (b) Trigger Number = 80     (c) Trigger Number = 160

(d) Trigger Number = 200     (e) Trigger Number = 320

Figure 17: Split on PubMed under different attack budget.

Table 8: ASR and Clean Accuracy before and after applying *LoSplit* under different trigger numbers.

| Dataset | Trigger Number | No Defense | | LoSplit | | | | |
|---|---|---|---|---|---|---|---|---|
| | | ASR ↓ | CA ↑ | ASR ↓ | CA ↑ | Prec. ↑ | Rec. ↑ | FPR ↓ |
| Cora | 10 | 100.0 | 82.96 | 0.00 | 84.44 | 100.0 | 100.0 | 0.00 |
| | 20 | 100.0 | 83.33 | 0.00 | 84.81 | 100.0 | 100.0 | 0.00 |
| | 40 | 100.00 | 81.85 | 0.00 | 85.19 | 100.0 | 100.0 | 0.00 |
| | 100 | 83.11 | 82.22 | 0.00 | 84.81 | 98.04 | 100.0 | 0.37 |
| | 160 | 92.89 | 80.74 | 0.00 | 84.07 | 98.16 | 100.0 | 0.55 |
| PubMed | 40 | 100.00 | 84.98 | 0.25 | 85.44 | 97.0 | 100.00 | 0.10 |
| | 80 | 92.82 | 84.93 | 0.42 | 85.19 | 65.57 | 100.00 | 1.07 |
| | 160 | 97.41 | 85.03 | 0.00 | 84.37 | 54.61 | 100.00 | 3.37 |
| | 200 | 85.14 | 85.19 | 0.08 | 85.19 | 71.10 | 100.00 | 2.05 |
| | 320 | 89.73 | 84.98 | 0.00 | 84.52 | 73.73 | 100.00 | 2.89 |

increases. The split effectively separate target nodes with perfect or perfect recall and near-perfect precision, while maintaining high clean accuracy (CA) above 84%.

On the PubMed dataset, the challenge becomes more pronounced as the trigger number increases from 80 to 320. The loss distributions exhibit greater overlap between target and clean nodes, leading to imperfect split. Despite this, *LoSplit* maintains remarkable robustness, reducing ASR from above 90% to almost zero while preserving clean accuracy around 84–85%. These results indicate that even under inaccurate identification scenarios, our proposed method in Decoupling-Forgetting Stage can still effectively suppress backdoor activation.

We also observe that achieving optimal performance on PubMed requires careful hyperparameter tuning across different trigger settings. The sensitivity of the Early Loss Dynamics to the data scale and the poisoning rate requires adjusting the split epoch $T_S$, the split learning rate $\eta_S$, and the balance coefficient $\gamma$. This observation highlights the importance of adaptive hyperparameter configuration in difficult-to-split scenarios, further validating the flexibility and stability of *LoSplit* under diverse attack intensities.

In summary, *LoSplit* exhibits strong robustness and adaptability against various attack budgets. Even in complex cases where target and clean nodes are not easily separable (e.g., PubMed with large trigger numbers), the defense remains highly effective with properly tuned parameters. Visualizations of split under different trigger numbers are shown in Fig. 16 and Fig. 17.

## J  Additional Results of Defense Performance under Different Backbone Model

To further evaluate the robustness of our method across different GNN architectures, we conduct experiments using both GAT and GraphSAGE as backbones. The results in Table 9 summarize the defense performance of *LoSplit* and RIGBD under various backdoor attacks (GTA, UGBA, DPGBA, and SPEAR) on three representative datasets (Cora, PubMed, and OGB-arXiv). Across all settings, *LoSplit* consistently achieves significantly lower ASR compared to both the original GAT and GraphSAGE, as well as the RIGBD defense. Moreover, the clean accuracy (ACC) of *LoSplit* remains comparable to or even higher than that of RIGBD, indicating that our decoupling-unlearning training does not sacrifice model utility. Notably, even under challenging attacks like SPEAR, which perturb only node features, *LoSplit* maintains ASR close to 0% and sustantially outperforms RIGBD, especially on the PubMed and OGB-arXiv datasets. These results demonstrate that *LoSplit* generalizes robustly across different GNN architectures. Overall, *LoSplit* is not tailored to any specific GNN design and can serve as a general-purpose training-time defense framework across various type of GNNs.

Table 9: Comparison of defenses under different attacks and different backbone models.

| Attack | Defense | Cora | | PubMed | | OGB-arXiv | |
|---|---|---|---|---|---|---|---|
| | | ASR(%) ↓ | CA(%) ↑ | ASR(%) ↓ | CA(%) ↑ | ASR(%) ↓ | CA(%) ↑ |
| GTA | GAT | 83.76 | 83.33 | 97.61 | 81.23 | 92.67 | 64.92 |
| | GraphSage | 99.63 | 84.07 | 98.73 | 84.68 | 90.83 | 64.75 |
| | RIGBD-GAT | 93.96 | 85.56 | 1.02 | 84.63 | 1.73 | 64.51 |
| | RIGBD-GraphSage | 99.63 | 78.89 | 98.97 | 75.14 | 80.24 | 64.47 |
| | LoSplit-GAT | 1.33 | 83.70 | 1.16 | 85.08 | 0.81 | 64.70 |
| | LoSplit-GraphSage | 2.58 | 80.74 | 0.58 | 84.98 | 0.59 | 64.30 |
| UGBA | GAT | 100.00 | 76.67 | 91.28 | 84.22 | 92.53 | 65.01 |
| | GraphSage | 97.33 | 84.07 | 95.38 | 86.25 | 96.16 | 65.66 |
| | RIGBD-GAT | 4.33 | 84.81 | 90.42 | 84.27 | 2.04 | 65.02 |
| | RIGBD-GraphSage | 98.16 | 77.04 | 90.11 | 82.30 | 2.06 | 65.75 |
| | LoSplit-GAT | 2.66 | 80.00 | 0.67 | 85.12 | 1.13 | 65.83 |
| | LoSplit-GraphSage | 4.00 | 82.96 | 1.67 | 83.51 | 0.74 | 65.95 |
| DPGBA | GAT | 100.00 | 83.70 | 91.53 | 84.07 | 92.78 | 66.13 |
| | GraphSage | 98.89 | 82.22 | 89.30 | 85.84 | 91.61 | 67.53 |
| | RIGBD-GAT | 6.22 | 83.33 | 0.84 | 83.51 | 1.28 | 64.86 |
| | RIGBD-GraphSage | 1.78 | 80.74 | 1.42 | 85.49 | 2.03 | 65.61 |
| | LoSplit-GAT | 0.89 | 84.07 | 0.51 | 83.90 | 1.02 | 65.31 |
| | LoSplit-GraphSage | 0.44 | 82.59 | 0.63 | 85.44 | 1.38 | 65.90 |
| SPEAR | GAT | 95.94 | 83.33 | 97.98 | 83.51 | 100.00 | 67.55 |
| | GraphSage | 100.00 | 84.44 | 83.00 | 85.59 | 95.00 | 67.00 |
| | RIGBD-GAT | 84.89 | 82.96 | 84.22 | 83.82 | 93.42 | 65.87 |
| | RIGBD-GraphSage | 100.00 | 82.96 | 100.00 | 85.24 | 91.56 | 65.99 |
| | LoSplit-GAT | 0.44 | 83.33 | 0.26 | 85.45 | 0.21 | 65.87 |
| | LoSplit-GraphSage | 0.37 | 83.33 | 0.00 | 85.64 | 0.00 | 66.09 |

# K   Training Algorithm

We summarize the training procedure of *LoSplit* for obtaining a backdoor-free GNN node classifier in Algorithm 1. Specifically, we begin by randomly initializing the parameters $\theta_{LoSplit}$ of an $L$-layer GNN $f_{LoSplit}$ that adopts the graph convolution operation defined in Eq. 1 (line 1). During the first training stage (lines 2–12), *LoSplit* trains $f_{LoSplit}$ on the backdoored graph $\mathcal{G}'_T$ using the RCE loss to capture early-stage loss dynamics. At each epoch $t \leq T_S$, *LoSplit* computes the loss per-node of RCE $\ell_i^{(t)}$ and evaluates its intra-class variance to identify the potential malicious label $y_t^{(t)}$ via Eq. (4) (lines 3–5) , and their loss values are standardized into z-scores (lines 6–8). A Gaussian Mixture Model (GMM) is then fitted to the z-scores to obtain two clusters, $\mathcal{C}_{\text{low}}^{(t)}$ and $\mathcal{C}_{\text{high}}^{(t)}$, representing clean and potentially poisoned nodes, respectively (line 9). *LoSplit* computes the separation metric $D^{(t)}$ between these clusters (lines 10–11) and selects the optimal split epoch $t^*$ that maximizes $D^{(t)}$ (line 13). Based on this epoch, the target and clean nodes are identified using Eq. 8 (line 14). In the second stage, *LoSplit* randomly initializes a new $L$-layer GNN $f$ and finetunes it using the decoupling objective defined in Eq. 9 to suppress the influence of backdoor nodes (lines 15–16). Finally, the algorithm outputs the backdoor-free GNN node classifier $f$ (line 17).

**Algorithm 1** Algorithm of *LoSplit*

---

**Require:** Backdoored graph $\mathcal{G}'_T = (\mathcal{V}'_T, \mathcal{A}'_T, \mathcal{E}'_T, \mathcal{X}'_T, \mathcal{Y}'_T)$; split epoch $T_S$, split learning rate $\eta_S$, Decoupling-Forgetting trade-off $\gamma$, RCE Constant $C = \log(1^{-10})$, Default threshold $\tau = 1^{-10}$
**Output:** Backdoor-free GNN node classifier
1: Randomly initialize $\theta_{LoSplit}$ for an L-layer GNN $f_{LoSplit}$ using Eq. 1;
2: **for** $t = 1, 2, ..., T_S$ **do**
3:    Compute RCE loss $\ell_i^{(t)}$ for all $v_i \in \mathcal{V}_T$;
4:    For each class $y_i \in \mathcal{Y}'_T$, compute intra-class variance of $\ell_i^{(t)}$;
5:    Malicious label $y_t$ identification via Eq. 4;
6:    Extract nodes with label $y_t$ as $\mathcal{V}_{y_t}^{(t)}$;
7:    Compute $\mu$, $\sigma$ of $\ell_i^{(t)}, \forall v_i \in \mathcal{V}_{y_t}^{(t)}$;
8:    Normalize loss into z-scores using Eq. 5;
9:    Fit GMM on z-scores to get clusters $\mathcal{C}_{\text{low}}^{(t)}, \mathcal{C}_{\text{high}}^{(t)}$;
10:   Compute threshold $\tau^{(t)}$ via Eq. 6;
11:   Compute $D^{(t)} = \mathbb{E}_{v_i \in \mathcal{C}_{\text{high}}^{(t)}}[\ell_i^{(t)}] - \mathbb{E}_{v_j \in \mathcal{C}_{\text{low}}^{(t)}}[\ell_j^{(t)}]$;
12: **end for**
13: Select optimal split epoch $t^* = \arg\max_t D^{(t)}$;
14: Split target and clean nodes via Eq. 8;
15: Randomly initialize $\theta$ for an L-layer GNN node classifier $f$;
16: Finetune $f$ via Eq. 9;
17: **Return** backdoor-free GNN node classifier $f$;

---

## L    Sample-wise Loss Distribution under RCE Loss in image and Comparison of Defense Strategies across Images and Graphs.

Fig. 18 illustrates the early-stage sample-wise loss distributions under the RCE loss on the CIFAR-10 dataset. In image domain, each poisoned example is an independent input with a localized trigger pattern (e.g., a small patch or pixel alteration) that directly dominates its prediction. As a result, the model can rapidly overfit to these triggers, leading to a highly stable loss distributionat the early training stage. This behavior reflects the strong spatial locality of visual backdoors and the independence among samples, where each image evolves in isolation during optimization. Such stability stands in sharp contrast to the graph domain discussed in introduction part. Unlike images, backdoor triggers in graphs are attached to a few interconnected nodes rather than independent samples, and their influence propagates *step-by-step* through message passing over limited hops. This structural dependency prevents the loss from converging smoothly, resulting in the unstable and oscillatory loss dynamics observed in Fig. 1. Therefore, while the fixed splitting ratio and global threshold is effective in images, directly transferring this intuition to graphs is not feasible due to the intrinsic interdependence and propagation characteristics of graph data. Table 10 briefly summarizes the difference between representative training-time defense in images and our proposed approach in graphs.

Table 10: Comparison of defense strategies across image and graph domains. "Splitting Ratio" indicates whether the partition between clean and poisoned samples is fixed or adaptively adjusted; "Threshold" denotes whether the separation criterion is globally fixed or adaptively determined during training.

| Domain | Method | Defense Strategy | Splitting Ratio | Threshold |
|--------|--------|------------------|-----------------|-----------|
| Image | ABL [18] | Unlearn | Fixed | – |
| Image | DBD [19] | Suppress | Fixed | – |
| Image | ASD [20] | Data Split | Fixed | Global |
| Image | HARVEY [22] | Data Split | Adaptive | Global |
| Graph | **LoSplit (Ours)** | Data Split | Adaptive | Adaptive |

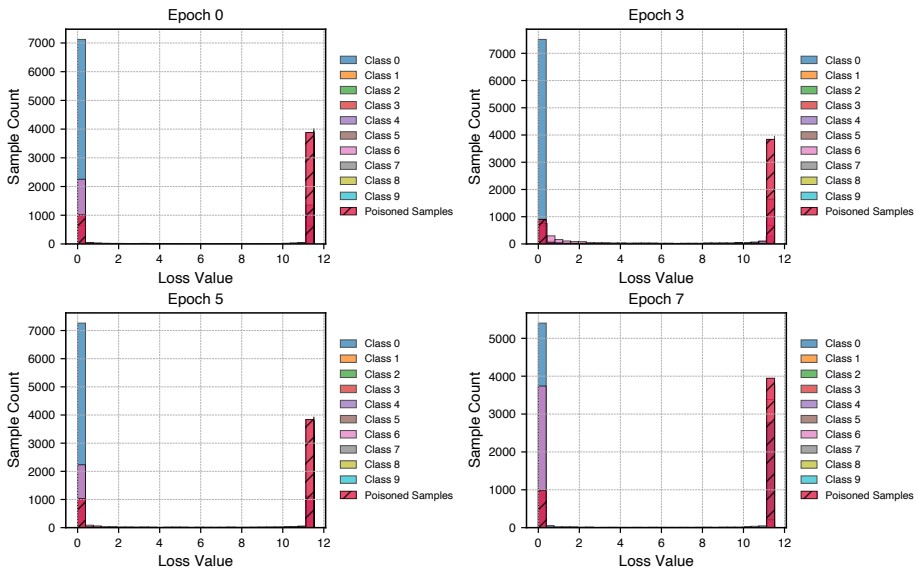

Figure 18: Distribution of the Early-Stage RCE Loss in Image Domain under Cifar-10 Dataset.

## M  Additional Results of Hyperparameter Analysis

To further evaluate the robustness of our method under the SPEAR attack, we conduct additional hyperparameter analyses on the PubMed and OGB-arXiv datasets. Following Sec. 5.5, we perform a grid search over $T_S \in \{3, 7, 11, 15, 19\}$, $\eta_S \in \{0.001, 0.005, 0.01, 0.05, 0.1\}$, and $\gamma \in \{0, 0.05, 0.1, \ldots, 1.0\}$.

For PubMed dataset, the hyperparameter analysis of *LoSplit* under the SPEAR attack is shown in Fig. 19. We observe that the ASR remains consistently near 0.0% across almost all configurations, demonstrating strong resistance to backdoor effects and the reliability of the split-based defense. The Precision peaks at 88.9% when $T_S = 19$ and $\eta_S \in \{0.005, 0.01\}$, while the Recall remains stable at 100%. These results suggest that a moderately late splitting epoch ($T_S = 19$) combined with a mid-range learning rate ($\eta_S = 0.005$ or $0.01$) yields the most balanced trade-off between precision and recall. Fixing $T_S = 19$ and $\eta_S = 0.005$, we further investigate the influence of the trade-off coefficient $\gamma$ in the Decoupling–Forgetting stage, which mainly governs the clean accuracy. As shown in Fig. 19d, the best clean accuracy of 85.13% is achieved at $\gamma = 0.05$, demonstrating that *LoSplit* effectively preserves model utility while maintaining low ASR. Overall, *LoSplit* exhibits stable and consistent performance on PubMed, maintaining high robustness and clean accuracy across a broad range of hyperparameters.

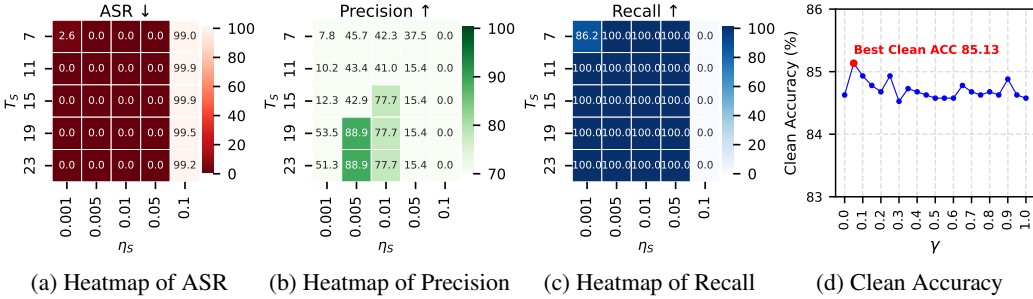

(a) Heatmap of ASR  (b) Heatmap of Precision  (c) Heatmap of Recall  (d) Clean Accuracy

Figure 19: Hyperparameter analysis of *LoSplit* on PubMed dataset under SPEAR attack.

For OGB-arXiv dataset, the results are presented in Fig. 20. We find that small splitting epochs ($T_S \leq 11$) and low learning rates ($\eta_S < 0.05$) significantly degrade the separation ability of *LoSplit*, leading to suboptimal splits. The highest Precision (up to 99.5%) and Recall (up to 96.8%) are achieved when $T_S \in \{19, 23\}$ and $\eta_S = 0.1$, suggesting that a sufficiently large learning rate with

later-stage splitting yields the most effective identification of target nodes. Fixing $T_S = 19$ and $\eta_S = 0.1$, we analyze the effect of $\gamma$ on clean accuracy. As illustrated in Fig. 20d, the optimal clean accuracy of 66.68% is achieved at $\gamma = 1.0$. These results confirm that *LoSplit* maintains strong robustness even on large-scale datasets like OGB-arXiv.

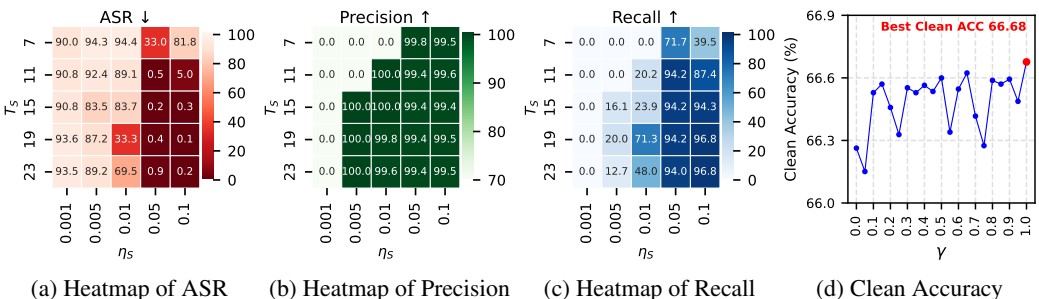

(a) Heatmap of ASR     (b) Heatmap of Precision     (c) Heatmap of Recall     (d) Clean Accuracy

Figure 20: Hyperparameter analysis of *LoSplit* on OGB-arXiv dataset under SPEAR attack.

# N    Reproducibility

Implementation and experimental details are illustrated in Appendix D and Sec. 5.1. We also provide a detailed training algorithm in Algo.1. The code for our *LoSplit* is publicly available at: github.com/zyx924768045/LoSplit.

# O    Limitations and Future Works

While *LoSplit* demonstrates strong capability in splitting target nodes and mitigating backdoor effects, its effectiveness may deteriorate when the distinction between target and clean nodes becomes less pronounced. In such cases, hyperparameters require careful adjustment through extensive trial and error, which can be both time-consuming and labor-intensive. Moreover, the performance of *LoSplit* can be sensitive to attack-specific parameters and training configurations (e.g. learning rate, hidden-layer dimensions), potentially limiting its robustness under diverse and real-world settings.

In future works, we plan to explore more self-adaptive defenses in both training-time and inference-time, and we hope to further enhance the practicality and robustness of backdoor defense across varying domains.

# P    Broader Impact

This paper presents work whose goal is improve the robustness of GNNs and thus advance the field of machine learning. While there may be potential societal consequences, we do not identify any that must be specifically highlighted here.

