# OpenReview forum: "LoSplit: Loss-Guided Dynamic Split for Training-Time Defense Against Graph Backdoor Attacks"
_NeurIPS.cc/2025/Conference — NeurIPS 2025 poster_

### Official Review · Reviewer_e9Mr · 2025-07-01

**Clarity:** 3
**Significance:** 3
**Originality:** 3
**Rating:** 5
**Confidence:** 4

**Summary:**

This paper proposes LoSplit, a novel training-time defense framework against both structure-based and feature-based graph backdoor attacks. Unlike prior methods that rely on detecting structural anomalies or outliers, LoSplit leverages early-stage loss dynamics. Experiments on multiple real-world datasets show that LoSplit significantly reduces attack success rates while maintaining high clean accuracy across diverse graph backdoor attacks.

**Questions:**

1. If attackers are aware of LoSplit’s defense mechanism, could they design triggers that intentionally delay loss convergence of target nodes? E.g. by weakening the correlation between the trigger and the target label?
2. What is the rationale behind using GMM for node separation in LoSplit? Given that the loss distributions may not always be strictly Gaussian, would non-parametric or density-based methods (e.g., DBSCAN or KDE) offer better robustness in challenging scenarios?

**Ethical Concerns:**

["NO or VERY MINOR ethics concerns only"]

**Final Justification:**

I have read the authors' careful responses, and most of my concerns have been addressed.

**Limitations:**

Yes.

**Paper Formatting Concerns:**

No.

**Quality:**

3

**Strengths And Weaknesses:**

**Strengths:**
1. Training-time backdoor defense for node classification is unexplored. LoSplit addresses this gap by focusing on early node-wise loss dynamics to defend against feature-based attacks, which existing methods fail to tackle.
2. The use of RCE loss and class-wise loss dynamics enables target node identification without relying on graph topology or outlier assumptions, enhancing robustness to stealthy perturbations.
3. LoSplit’s Decoupling-Unlearning strategy breaks associations between target nodes and malicious labels without degrading clean accuracy, showing significant performance improvement over the robust training strategy mentioned in RIGBD.

**Weaknesses:**
1. LoSplit retains sensitivity to training configurations and specific attack parameters, which could hinder its robustness and generalizability in diverse or unpredictable real-world settings.
2. The Decoupling-Unlearning strategy randomly relabels target nodes to classes other than target labels, which fails to restore their original labels (before attack). This may inadvertently sacrifice a little clean accuracy by misassigning original labels.
3. LoSplit relies on clear loss separation within the target class. When the distributions of target and clean nodes within the target class heavily overlap, the accuracy of GMM-based separation may degrade.
4. The clean-label setting is well motivated, but the paper could benefit from a more direct comparison (qualitative or conceptual) with label-flipping approaches to highlight its practical trade-offs.

---

> ### Author Rebuttal · Authors · 2025-07-29
>
> Rebuttal to Reviewer e9Mr
>
> **W1.** LoSplit retains sensitivity to training configurations and specific attack parameters, which could hinder its robustness and generalizability in diverse or unpredictable real-world settings.
>
> #### **Response:**
> >We appreciate the reviewer’s concern regarding certain robustness. While we respectfully acknowledge that LoSplit may be sensitive to certain model training and attack configurations, we have explicitly discussed this limitation in the paper (see Appendix M). We have designed it with minimal reliance on attack-specific knowledge. In our experiments, we consistently use the same configuration across different datasets, and LoSplit remains effective even under varied settings.  As noted in limitations, improving the robustness and generalizability of LoSplit under diverse or unpredictable real-world settings is an insightful future direction.
>
>  **W2.** The Decoupling-Unlearning strategy randomly relabels target nodes to classes other than target labels, which fails to restore their original labels (before attack). This may inadvertently sacrifice a little clean accuracy by misassigning original labels.
>
> #### **Response:**
> >Thanks for your constructive feedback. In response, we experimented with manually correcting the labels of target nodes to their original clean labels. However, this adjustment resulted in only **marginal gains** in clean accuracy, indicating limited practical benefit. Moreover, automatic label correction methods may not always accurately recover to the original labels but often **introduce additional computational cost**. Considering these trade-offs, we believe our combination of random relabeling is a more effective and efficient strategy to suppress backdoor effect while maintaining model robustness.
>
> **W3.** LoSplit relies on clear loss separation within the target class. When the distributions of target and clean nodes within the target class heavily overlap, the accuracy of GMM-based separation may degrade.
>
> #### **Response:**
> >Thank you for the insightful comment. We agree that the performance of GMM-based separation may degrade when the loss distributions of poisoned and clean nodes within the target class are overlapping. To investigate this, we have conducted experiments in such cases. As shown in **Figure 12–17** in **Appendix**, even when the two loss distributions are close or even overlap, **LoSplit** maintains effective, owing to its adaptive thresholding and exploitation of early-stage loss dynamics. We will clarify this point in the revised version.
>
> **W4.** The clean-label setting is well motivated, but the paper could benefit from a more direct comparison (qualitative or conceptual) with label-flipping approaches to highlight its practical trade-offs.
>
> #### **Response:**
> >We honestly apologize for the confusion. We realize that this point may not have been clearly stated in the current version. In fact, the attacks we consider **all involve flipping the labels of target nodes**,  **not clean-label setting**, meaning that our setting already **aligns with the label-flipping paradigm** (dirty-label attack). We will clarify the label settings in the revised version.
>
>  **Q1.** If attackers are aware of LoSplit’s defense mechanism, could they design triggers that intentionally delay loss convergence of target nodes? E.g. by weakening the correlation between the trigger and the target label?
>
> #### **Response:**
> >We sincerely appreciate the reviewer’s concern and agree that evaluating adaptive attacks is important. Our work focuses on established graph backdoor attacks (GTA, UGBA, DPGBA, SPEAR), where early and sharp loss convergence of poisoned nodes is a consistent and empirically verified property (see Section 4.1 and Figure 2), and we further provide a theoretical explanation in Appendix C.1. To the best of our knowledge, such a threat model has not been considered in existing graph backdoor attack literature. If an attacker intentionally suppresses this convergence, e.g., by forcing poisoned nodes to mimic clean node loss trajectories, it would largely weaken the backdoor effect. In such a case,  it deviates the essence of backdoor attacks. While this scenario diverges from current threat models, we agree it is a compelling direction for future research on both adaptive attack strategies and more resilient defenses.
>
> **Q2.** What is the rationale behind using GMM for node separation in LoSplit? Given that the loss distributions may not always be strictly Gaussian, would non-parametric or density-based methods (e.g., DBSCAN or KDE) offer better robustness in challenging scenarios?
>
> #### **Response:**
> >We truly appreciate the reviewer's question regarding the rationale for using GMM in LoSplit. We chose it for its applicability, simplicity and empirical effectiveness in modeling the early-stage loss distributions of nodes within a class. While we respectfully acknowledge that the underlying distributions may not be perfectly Gaussian, in practice we observe that the loss distribution shape between poisoned and clean nodes within the same target class tends to induce approximate bimodal Gaussian distribution, making GMM a reasonable and efficient choice. To further validate this design, we have experimented  alternatives such as DBSCAN and kernel density estimation (KDE). However, these methods either produced less stable clustering results or showed inferior performance in distinguishing target nodes across different datasets. Based on these findings, we opted to retain GMM as the most reliable and interpretable solution.

---

> > ### Comment · Reviewer_e9Mr · 2025-08-06
> >
> > I have read the authors' careful responses, and most of my concerns have been addressed.

---

> ### Author Response · Authors · 2025-08-06
> **Official Comment by Authors**
>
> We sincerely appreciate your careful reading of our responses. We’re pleased that our clarifications helped resolve most of your concerns, and we are grateful for your thoughtful and constructive review.

---

### Official Review · Reviewer_F71D · 2025-07-01

**Clarity:** 3
**Significance:** 3
**Originality:** 3
**Rating:** 5
**Confidence:** 3

**Summary:**

The paper looks at the problem of backdoors in graph node classification tasks. Taking inspiration from defenses such as ABL, the paper proposes an identification phase that uses the loss separation between the nodes at the best possible epoch (using a particular best epoch selection scheme). These identified backdoored nodes are then unlearned using a loss function that maximizes the cross-entropy w.r.t. the label in the dataset while minimizing the cross-entropy w.r.t. a randomly chosen target (while using a CE loss on the clean nodes to retain performance). The authors evaluated their proposed defense methodology on four node classification datasets, and using four different attack methods.

**Questions:**

- Why not bold the best result in table 2?
- Page 2 (contributions): RCE was introduced without the full term which was introduced on page 4. Maybe rewrite to remove RCE term here as it doesn't really add anything.

**Ethical Concerns:**

["NO or VERY MINOR ethics concerns only"]

**Final Justification:**

The authors demonstrated that their backdoor identification method is significantly stronger than ABL, which is convincing. I am not convinced that their unlearning approach is the best. I think the paper has interesting contributions. Would have been a good contribution if they were able to tease apart the identification from unlearning (which they partially attempted to do during the rebuttal), but I still believe that this would be a useful reference point for further work in this space in the future. Hence, I updated my score to an accept.

**Limitations:**

yes

**Quality:**

3

**Strengths And Weaknesses:**

# Strengths

- Important problem
- Well-written paper
- Comprehensive experiments

# Weaknesses

- **Other identification techniques not included:** it is sensible to argue that ABL is a heuristic, specifically the epoch selection part. The authors carefully establish the epoch selection scheme. I am wondering why the authors didn't attempt to compare against other approaches that were proposed to get rid of this epoch selection heuristic in ABL (e.g., see https://openreview.net/forum?id=rK0YJwL69S).
- **Unclear unlearning scheme:** I am not sure if I see the utility of two different unlearning methods being integrated i.e., gradient ascent as well as the random target learning (eq. 9) -- these weren't ablated directly. It would be important to see if these two add something together, or if one of these terms is already sufficient. I am also wondering if the random learning was a missed opportunity where the target label could have been computed more carefully following the label correction literature (e.g., see https://arxiv.org/abs/1904.11238)
- **Captions for tables/figures provide no takeaways:** given the current pace of the field, many readers just go through the figures and tables in the paper without reading the text. The current captions provide no useful information independently. I would suggest the authors to rewrite the captions such that they are independent from the text so that one can understand the paper by just going through these figures/tables independently like all recent well-written papers do.

---

> ### Author Rebuttal · Authors · 2025-07-29
>
> ## Rebuttal to Reviewer F71D
>
> **W1.** Other identification techniques not included: it is sensible to argue that ABL is a heuristic, specifically the epoch selection part. The authors carefully establish the epoch selection scheme. I am wondering why the authors didn't attempt to compare against other approaches that were proposed to get rid of this epoch selection heuristic in ABL (e.g., see [2]).
>
> #### **Response:**
>
> >Thank you for your insightful comment regarding the heuristic epoch selection of ABL [3].  We are aware of recent work, particularly BaDLoss [2], which replaces the fixed-epoch loss heuristic with a trajectory-based anomaly score, comparing each sample’s entire loss curve to a reference set of clean examples.
>
> >We did not compare against BaDLoss explicitly for the following reasons: **(1)** BaDLoss depends on access to a known clean subset of the training data (e.g., 250 clean samples in their experiments) to compute trajectory-based anomaly scores. Our method assumes no such clean set is available, because this assumption is not applicable in the graph setting. In **image domains, clean samples are often easily recognized**, triggers typically appear as visible patterns such as black squares or watermarks on the image, making manually selected clean set feasible. In contrast, **trigger nodes in graphs are visually indistinguishable from clean nodes**, and their effects are only revealed indirectly through message passing. This makes it impractical to access to any bona fide clean subset in graph-based tasks beforehand. *(2)** BaDLoss addresses a different threat model: BaDLoss is primarily designed for multi-attack settings in image domain, where multiple simultaneous poisoning attacks are present. Our setting assumes a single backdoor attack at a time, aligning with ABL. Direct comparison would require adapting BaDLoss to our threat model and hyperparameter setup (e.g., clean subset, trajectory depth), which is nontrivial and beyond the current scope.**(3)** As the BaDLoss authors themselves note (Appendix B), ABL can be interpreted as a special case of BaDLoss using a single point in the loss trajectory. Hence, our inclusion of ABL already implicitly establishes a comparison baseline for BaDLoss kind of methods.
>
>
>  **W2.** Unclear unlearning scheme: I am not sure if I see the utility of two different unlearning methods being integrated i.e., gradient ascent as well as the random target learning (eq. 9) -- these weren't ablated directly. It would be important to see if these two add something together, or if one of these terms is already sufficient. I am also wondering if the random learning was a missed opportunity where the target label could have been computed more carefully following the label correction literature (e.g., see [1])
>
> #### **Response:**
> >We would like to thank you for your valuable comment. We agree with the reviewer that either gradient ascent or random target learning can contribute to mitigating backdoor effects theoretically. However, in our experiments, **gradient ascent alone was ineffective** in reducing the attack success rate due to **gradient explosion**. To address this, we introduce **random target learning** as a complementary mechanism that provides more stable training process for unlearning. What is **worth mention** is that our **random label assignment is different in every training iteration**, meaning that each target node learns a different random label at each epoch. This introduces versatile perturbation to the label learning of target nodes and prevents the model from overfitting any fixed incorrect label.
>
> >To support our design choice, we conducted an **ablation study under the DPGBA attack on the Pubmed dataset**, comparing each component in isolation and their combination. As shown below, while random label assignment alone achieves reasonable defense performance, combining it with gradient ascent further **suppresses the attack success rate (ASR)** while maintaining clean accuracy (ACC).
>
> **Table 1: Ablation study on unlearning components under DPGBA attack (Pubmed).**
> *The combined use of gradient ascent and random target learning yields the best trade-off between attack success rate (ASR) and clean accuracy (ACC).*
>
> | Method                | ASR ↓ | ACC ↑ |
> |----------------------|--------|--------|
> | Gradient Ascent only | 98.19   | 84.78  |
> | Random Label only    | 1.04    | 85.08  |
> | Both (Ours)          | **0.98**    | **85.13**  |
>
> >These results validate that the combination provides complementary benefits: **gradient ascent targets the removal of memorized backdoor patterns**, while **random label learning helps prevent overfitting and stabilizes training**. Together, they achieve stronger unlearning performance than either component alone. We will revise our paper to further clarify the rationale for combining both techniques, and (2) include the ablation study above in Appendix.
>
> >In response to the reviewer’s suggestion on using **label correction** methods [1], we have manually corrected the labels of the target nodes to match their original clean labels. However, we observed that this brought **negligible improvement in clean accuracy** , let alone some label correction methods may occasionally not recover to their original labels but add additional computation cost. Given these, we believe both our random label assignment is better than label correction methods.
>
> **W3.** Captions for tables/figures provide no takeaways: given the current pace of the field, many readers just go through the figures and tables in the paper without reading the text. The current captions provide no useful information independently. I would suggest the authors to rewrite the captions such that they are independent from the text so that one can understand the paper by just going through these figures/tables independently like all recent well-written papers do.
>
> #### **Response:**
> >We fully agree with this suggestion. Clear and self-contained captions significantly improve readability. In the revision, we will rewrite all figure and table captions to (i) include the key takeaway message and (ii) briefly describe the experimental setting where necessary, so that readers can interpret the results without referring to the main text.
>
>
> **Q1.** Why not bold the best result in table 2?
>
> #### **Response:**
> >Thank you for pointing this out. The omission of bolded best results in Table 2 (precision, recall, and FPR comparison) was an oversight on our part, and we sincerely apologize for the inconsistency. In the revised version, we will correct this by clearly bolding the best-performing results for each metric and ensure consistent formatting across all evaluation tables. We will also revise the caption to better summarize the key takeaway regarding identification effectiveness.
>
>
> **Q2.** Page 2 (contributions): RCE was introduced without the full term which was introduced on page 4. Maybe rewrite to remove RCE term here as it doesn't really add anything.
>
> #### **Response:**
> >We respectfully appreciate the reviewer’s feedback. We agree that introducing the acronym “RCE” before defining it may be confusing and we sincerely apologize for our carelessness. However, we believe the use of Reverse Cross Entropy (RCE) is an important contribution of our method as it significantly amplifies the rate of loss drop differences between clean and poisoned nodes during early training, which is crucial for effective identification. We have included a theoretical justification in the Appendix C.1 to support why RCE is more effective than standard Cross Entropy (CE) in this context. In the revised version, we will provide the full name before mention its acronym.
>
>
> #### **Reference**
> [1] Unsupervised Label Noise Modeling and Loss Correction. ICML, 2019.
>
> [2] Protecting against simultaneous data poisoning attacks. ICLR, 2025.
>
> [3] Anti-Backdoor Learning: Training Clean Models on Poisoned Data. NeurIPS, 2021.

---

> > ### Comment · Reviewer_F71D · 2025-08-05
> > **Official comment from reviewer F71D**
> >
> > I am very thankful to the authors for their detailed response.
> > (i) I don't care about BaDLoss in particular. All I am concerned about is the arbitrary epoch selection criterion for ABL (i.e., the identification part), which is the biggest weakness of that approach (despite using a reasonable unlearning algorithm i.e., gradient-ascent).
> > (ii) I am happy to see that ablation which was in general quite important to understand the contributions of the paper. Correct me if I'm wrong, but seems like random label is already sufficient, and using both doesn't add much according to your plot? I am quite well-aware about the limitations of random relearning in regards to relearning susceptibility, but just focusing on your results for now. Why should we use both then and complicate the training setup?
> > (iii) I wasn't pushing for label correction. I was just wondering whether that's a possibility given that the authors ran those experiments.
> >
> > I know the time is limited (apologies), but I am just wondering if there is any possibility to compare different identification approaches and different unlearning approaches, which I think would significantly strengthen the contribution of this work. I would feel much more comfortable pushing for its acceptance in that case. Otherwise, everything else as of now is a straight-forward adaptation of existing work (e.g., see https://arxiv.org/abs/2402.14015).

---

> ### Author Response · Authors · 2025-08-06
> **Responses to reviewer F71D (Part 1)**
>
> *We sincerely thank you for your insightful follow-up. Below, we will provide detailed and thorough responses to your additional questions.*
>
> (i) I don't care about BaDLoss in particular. All I am concerned about is the arbitrary epoch selection criterion for ABL (i.e., the identification part), which is the biggest weakness of that approach (despite using a reasonable unlearning algorithm i.e., gradient-ascent).
>
> **Response:**
>
> >We’re more than happy to hear that our explanation helped clarify this aspect. As you correctly pointed out the weakness of ABL. Our design of **LoSplit** was just tailored to solve this weakness: instead of fixing a random early epoch, we dynamically select the most discriminative one.
>
> (ii) I am happy to see that ablation which was in general quite important to understand the contributions of the paper. Correct me if I'm wrong, but seems like random label is already sufficient, and using both doesn't add much according to your plot? I am quite well-aware about the limitations of random relearning in regards to relearning susceptibility, but just focusing on your results for now. Why should we use both then and complicate the training setup?
>
> **Response:**
>
> >We agree that **random relabeling alone performs well in most cases**. However, we suspect it might have **limitations** that could sacrifice its effectiveness under certain scenario.
>
> > Specifically, during training, we suggest that **random relabeling would gradually cause the model to converge to an "average state"**, where the output of the target node approximates a **uniform distribution across classes** (i.e., predicting all classes equally).
>
> > Although this average convergence can suppress the learning of any malicious labels, we **speculate that in some cases**, this "average state" could **accidentally align with the state of certain target classes**, where the representation lies closely to the center of the embedding space (average state).
>
> > To support this speculation, we **visualized the learned embeddings using t-SNE**, and observed that **classes 0 and 1 consistently lie near the center of the embedding space**, whereas **classes 2 and 4 are farther away**.  Interestingly, this aligns with our results: **target classes 0 and 1 tend to achieve the lowest clean accuracy**, while **target classes 2 and 4 consistently yield the highest**.  Although t-SNE only offers an indirect low-dimensional projection, this observation provides  **potential explanations** at some points.
>
> >This implies that the use of **gradient ascent as a complementary** would play a vital role in **pushing the node’s representation away** from the malicious label, which could help the unlearning process more directional, potentially mitigating the risk of convergence toward the centroids of the embedding space. Therefore, while **random relabeling is effective on most cases**, we believe that **combine it with gradient ascent will be more robust**.
>
> >To validate this, we have conducted experiments with **tunable weights (α for random relabeling (R), β for gradient ascent (GA) )**. Below are results on DPGBA (Cora), across different target classes:
>
> #### **Table 1:** ASR ↓ and ACC ↑ Comparison across Target Classes (DPGBA Attack on Cora)
> | Target Class | Method            | ASR ↓  | ACC ↑   | α (R) | β (GA) |
> |--------------|-------------------|--------|---------|--------|--------|
> | 0            | R                 | 0.00   | 84.44   | 1.00   | 0.00   |
> |              | GA                | *10.22*  | 84.07   | 0.00   | 1.00   |
> |              | Both              | 0.00   | 84.44   | 1.00   | 1.00   |
> |              | Both (Tuned)      | 0.00   | **84.81** | 0.60   | 0.40   |
> | 2            | R                 | 0.00   | 85.56   | 1.00   | 0.00   |
> |              | GA                | 0.00   | 85.19   | 0.00   | 1.00   |
> |              | Both              | 0.00   | 85.56   | 1.00   | 1.00   |
> |              | Both (Tuned)      | 0.00   | **85.93** | 0.30   | 0.70   |
> | 6            | R                 | 0.00   | 83.33   | 1.00   | 0.00   |
> |              | GA                | 0.00   | 84.44   | 0.00   | 1.00   |
> |              | Both              | 0.00   | 83.33   | 1.00   | 1.00   |
> |              | Both (Tuned)      | 0.40   | **84.81** | 0.40   | 0.60   |
>
> ---
> > **Key Takeaway:** We empirically verified that the combination of **Gradient Ascent** and **Random Relabeling** can truly improve clean accuracy by carefully tuning the trade-offs. This suggests that both loss components, weighted by α and β, play a meaningful role in model relearning. Notably, in **Target Class 0**, using gradient ascent alone led to a **significant deterioration in ASR (10.22)**, revealing the instability of GA under certain class conditions.
>
> >Based on current evidence, we find both component are important, and we will add α, β explicitly in Eq. (9) and include a hyperparameter analysis in the revised paper to further investigate their impact.

---

> > ### Comment · Reviewer_F71D · 2025-08-08
> > **Contribution of loss terms**
> >
> > I am thankful to the authors for highlighting the contributions further. I think the gains in clean accuracy seem quite small to be concerned about, no?
> >
> > It seems to me that instead of using these combinations in an ad-hoc manner, it would have been useful to just use something like the SCRUB framework for unlearning (https://arxiv.org/abs/2302.09880), which minimizes the KL-div and CE on the retain set, and maximizes the KL-div on the forget set (which would be equivalent to the identified backdoored examples in your case). I do understand the authors' perspective that others would have complained about novelty in that case, but it seems to be scientifically better to reuse where possible, unless there is a good reason to deviate, which I am finding it difficult to see.

---

> ### Author Response · Authors · 2025-08-06
> **Responses to reviewer F71D (Part 2)**
>
> (iii) I wasn't pushing for label correction. I was just wondering whether that's a possibility given that the authors ran those experiments.
>
> **Response:**
> >You’re right in some point that label correction is a valid alternative that we considered early on. However, in graph domains, label correction is much more expensive and time-consuming than in images:
>
> >**In images**, poisoned samples are often **easily distinguishable by human eyes (eg., a dog with a strange patch)**, making manual relabeling feasible and light-weight.
>
> >**In graphs**, however, a node’s label depends not only on its features, but also on its neighborhood and topology, making it non-trivial to determine its ground truth. Thus, label correctness in graphs is more laborious. Our defense is to develop an **automated**  method, avoiding **human-in-the-loop** mechanisms whenever possible.
>
> >To further support our choice of random relabeling over manual correction, we conducted experiments comparing performance using the **original labels (O)** and **random relabeling (R)** across four attack settings on the Cora dataset:
>
> #### **Table2: Original Label vs. Random Relabeling under Four Attacks (Cora Dataset)**
>
> | Attack | Type | ASR ↓ | ACC ↑ |
> |--------|------------|--------|----------------|
> | GTA    | O   | 0.00   | 83.33          |
> |        | R     | 0.00   | **83.70**      |
> | UGBA   | O   | 0.00   | **85.19**      |
> |        | R     | 0.00   | 84.81          |
> | DPGBA  | O   | 1.78   | 84.44          |
> |        | R     | **0.00**   | **84.81**      |
> | SPEAR  | O   | 0.00   | **84.07**      |
> |        | R     | 0.00   | 83.70          |
>
> >**Key Takeaway:** **Random Relabeling performs on par or even better than using the original labels**, while avoiding any reliance on manual relabeling. This aligns with our goal of building a **fully automated, human-free defense pipeline** for graph domains. Notably,  **UGBA and SPEAR** using original labels yields only **marginal accuracy gains**, further validating the utility of our relabeling strategy.
>
> I know the time is limited (apologies), but I am just wondering if there is any possibility to compare different identification approaches and different unlearning approaches, which I think would significantly strengthen the contribution of this work. I would feel much more comfortable pushing for its acceptance in that case. Otherwise, everything else as of now is a straight-forward adaptation of existing work.
>
> **Response:**
>
> >Regarding your question about whether we could compare different *identification* and *unlearning* approaches, we fully agree that such an ablation would further strengthen the contribution of our method. To this end, we add an experiment comparing the combinations of:
>
> - **Identification (ID) Methods:**
>
> >RIGBD [1] (RED (Random Edge Dropping) )
>
> > OD [2] (Outlier Detection)
>
>  >ABL (CE Loss)
>
> > **LoSplit (RCE Loss)**
>
> - **Unlearning (UL) Methods:**
>
>  >RIGBD ( RT (Robust Training))
>
>  >ABL ( GA (Gradient Ascent))
>
>  >**LoSplit (GA+R (Gradient Ascent + Random Relabeling))**
>
> **Table 3:** ASR ↓ / ACC ↑ comparison using different combinations of ID and UL methods.
> *Experiments are conducted on the **Pubmed** dataset under the **SPEAR** attack.*
>
> | ID ↓ / UL →       | RIGBD (RT) | ABL (GA) | LoSplit (GA+R) |
> |-------------------|------------|----------|----------------|
> | **RIGBD (RED)**   | 92.65/83.51 | 90.32/84.88 | 90.19/84.99 |
> | **OD**            | 0.17/63.27  | 28.21/84.93 | 2.75/75.49  |
> | **ABL (CE)**      | 0.92/85.29  | 0.42/84.83  | 0.28/84.88 |
> | **LoSplit (RCE)** | 0.08/84.68  | 100.00/76.36 | **0.08/84.98** |
>
> >**Key Takeaway:** Our LoSplit (both ID and UL method) consistently achieve best results across prior works **especially SOTA method RIGBD**. This demonstrates that our method is **not a straightforward adaptation**, but a carefully designed and empirically validated approach. We will include this table in the revised version.
>
> >**Due to time limits**, we focused on combinations of existing methods applied to graphs. A broader comparison with techniques from other domains (e.g. vision) is a promising future direction that we plan to explore.
>
> >*We truly appreciate your continued engagement with the paper, and we are very glad to respond to your further questions promptly and thoroughly.*
>
> **Reference**
>
> [1] Robustness Inspired Graph Backdoor Defense, ICLR 2025.
>
> [2] Rethinking Graph Backdoor Attacks: A Distribution-Preserving Perspective, KDD 2024.

---

> > ### Comment · Reviewer_F71D · 2025-08-08
> > **Trying to understand table 3**
> >
> > Apologies for the late response. I am thankful to the authors for running these additional experiments on such short notice. I think doing this kind of analysis, such as the one presented in Table 3, is very useful to understand the true contribution.
> >
> > I am reading it correctly that the best identification method is ABL (all unlearning methods seem to play well with ABL identification)? I don't really find the differences below 1% to be meaningful. These results seem a little unsettling, considering that the story of the paper is around the fact that we should get beyond ABL for identification due to an arbitrary identification threshold computed at an arbitrary epoch that is not readily translatable to new settings.
> >
> > The fact that LoSplit identification coupled with ABL unlearning achieved an ASR of 100% while ABL coupled with any unlearning algorithm achieved an ASR of <= 1% indicates that ABL is the most effective identification technique in this current evaluation setup, and achieves the best resistance against attacks when coupled with any unlearning algorithm.
> >
> > Happy to hear any clarifications on this from the authors' side, but appreciate them running this experiment.

---

> ### Author Response · Authors · 2025-08-09
> **Responses to reviewer F71D (Trying to understand table 3)**
>
> > We appreciate your raising this concern and we would like to clarify the confusion about LoSplit’s superiority over ABL.
>
> >First of all, the poor performance in the middle of Table 3’s bottom line (100.00 / 76.36) was **not due to the identification method itself**. It was caused by an implementation oversight, we missed a ReLU function in the gradient ascent step to stabilize training which is **inconsistent with ABL (ID) + ABL (UL)**. We sincerely apologize for this oversight due to tight time schedule. After adding the ReLU function, the revised result for (LoSplit + ABL) is: **0.41/84.93**.
>
> > Moreover, there is a **big difference in the process of ABL and LoSplit** method. ABL uses a **single model** (early epochs for identification, later for unlearning). In contrast, LoSplit employs **two separate models**: one is naively trained for identification, the other is fully trained for unlearning. This fundamental difference may cause **compatibility issues**.
>
> > For pure identification ability comparison, we use three metrics to evaluate:
>
> **Table 4.** Identification performance:
>
> | Method   | Precision↑  | Recall↑   | FPR↓   |
> |----------|-------------|-----------|--------|
> | **ABL**      | 97.56       | 25.00     | 0.03   |
> | **LoSplit**  | 90.91       | 100.00    | 0.41   |
>
> > **Precision**: proportion of detected nodes truly poisoned;
> >**Recall**: proportion of truly poisoned nodes correctly identified;
> >**FPR**: proportion of clean nodes wrongly categorized.
>
> > ABL has slightly higher precision and lower FPR but very poor recall, missing many poisoned nodes. LoSplit has perfect recall and slightly higher FPR, showing superior overall detection.
>
> > As evaluation was on one attack and dataset, results may reflect some randomness and limited generalizability. To further prove LoSplit’s superiority, we provide results on two attacks (**UGBA**, **DPGBA**) and two datasets (**Cora**, **Pubmed**):
>
> **Table 5:** UGBA (Pubmed)
>
> | ID + UL           | ASR ↓  | ACC ↑  | Precision ↑ | Recall ↑ | FPR ↓  |
> |-------------------|--------|--------|-------------|----------|--------|
> | LoSplit + LoSplit  | 0.26   | **85.34**  |  100.00      | **100.00**   | 0.00   |
> | LoSplit + RIGBD    | 0.45   | **84.53**  | —           | —        | —      |
> | LoSplit + ABL      | **89.38**  | **84.93**  | —           | —        | —      |
> | ABL + LoSplit      | **0.13**  | *83.51*  | 100.00      | *25.62*    | 0.00   |
> | ABL + RIGBD        | **0.32**   | *84.22*  | —           | —        | —      |
> | ABL + ABL          | 98.78  | 84.42  | —           | —        | —      |
>
> ---
>
> **Table 6:** UGBA (Cora)
>
> | ID + UL           | ASR ↓  | ACC ↑  | Precision ↑ | Recall ↑ | FPR ↓  |
> |-------------------|--------|--------|-------------|----------|--------|
> | LoSplit + LoSplit  | **0.44**   | **83.33**  | 97.56       | **100.00**   | 0.18   |
> | LoSplit + RIGBD    | **12.00**  | **81.11**  | —           | —        | —      |
> | LoSplit + ABL      | **11.11**  | **83.70**  | —           | —        | —      |
> | ABL + LoSplit      | 73.33  | 48.14  | 100.00      | *16.25*    | 0.00   |
> | ABL + RIGBD        | 76.00  | 79.63  | —           | —        | —      |
> | ABL + ABL          | 95.56  | 79.63  | —           | —        | —      |
>
> ---
>
> **Table 7:** DPGBA (Pubmed)
>
> | ID + UL           | ASR ↓  | ACC ↑  | Precision ↑ | Recall ↑ | FPR ↓  |
> |-------------------|--------|--------|-------------|----------|--------|
> | LoSplit + LoSplit  | 1.61   | **84.73**  | 93.96       | **87.50**    | 0.23   |
> | LoSplit + RIGBD    | **0.90**   | 84.07  | —           | —        | —      |
> | LoSplit + ABL      | **1.09**  | **85.29**  | —           | —        | —      |
> | ABL + LoSplit      | **0.45**  | *83.96*  | 100.00      | *25.26*    | 0.00   |
> | ABL + RIGBD        | 0.90   | **84.42**  | —           | —        | —      |
> | ABL + ABL          | 1.48   | 83.51  | —           | —        | —      |
>
> ---
>
> **Table 8:** DPGBA (Cora)
>
> | ID + UL           | ASR ↓  | ACC ↑  | Precision ↑ | Recall ↑ | FPR ↓  |
> |-------------------|--------|--------|-------------|----------|--------|
> | LoSplit + LoSplit  | **0.00**   | **84.81**  | 95.24       | **100.00**   | 0.37   |
> | LoSplit + RIGBD    | **0.89**   | 81.85  | —           | —        | —      |
> | LoSplit + ABL      | **10.22**  | **84.07**  | —           | —        | —      |
> | ABL + LoSplit      | 10.22  | 80.37  | 100.00      | *12.50*    | 0.00   |
> | ABL + RIGBD        | 11.11  | **83.70**  | —           | —        | —      |
> | ABL + ABL          | 49.78  | 80.37  | —           | —        | —      |
>
> ---
>
> > **Key Takeaway:** LoSplit identification consistently achieves the lowest ASR with **stable unlearning (LoSplit and RIGBD)** and far better identification ability over ABL. **ABL performs poorly, especially with Cora under UGBA attack**. The result (**ABL + LoSplit: 73.33/48.14**) reveals **compatibility issues**. This further shows the prior gap was due to compatibility issues and unlearning settings.

---

> ### Author Response · Authors · 2025-08-09
> **Responses to reviewer F71D (Contribution of loss terms)**
>
> I am thankful to the authors for highlighting the contributions further. I think the gains in clean accuracy seem quite small to be concerned about, no?
>
> **Response:**
>
> >We respectfully argue with the reviewer’s statement that while the present results show slight gain in clean accuracy (around 0.4% plus compared to one term alone), we do believe **for different attacks, datasets, and target classes, there always exists an optimal combination of α and β (by hyperparameter analysis) that can lead to a substantial boost in clean accuracy**. And we have already find such cases. To validate this, we run the SPEAR attack on the Cora dataset with target class 2 and 3:
>
> **Table 9:** ASR ↓ and ACC ↑ Comparison across Target Classes (SPEAR Attack on Cora)
>
> | Target Class | Method        | ASR ↓ | ACC ↑   | α     | β     |
> |--------------|--------------|-------|---------|-------|-------|
> | 2            | R            | 0.00  | 82.59   | 1.00  | 0.00  |
> |              | GA           | 0.00  | 83.33   | 0.00  | 1.00  |
> |              | Both         | 0.00  | 82.59   | 1.00  | 1.00  |
> |              | Both (Tuned) | 0.00  | **85.56** | 0.15  | 0.85  |
> | 3            | R            | 0.00  | 82.59   | 1.00  | 0.00  |
> |              | GA           | 0.00  | 83.33   | 0.00  | 1.00  |
> |              | Both         | 0.00  | 82.59   | 1.00  | 1.00  |
> |              | Both (Tuned) | 0.00  | **84.81** | 0.25  | 0.75  |
>
> > **Key Takeaway:** These results demonstrate that for different target classes, careful tuning of α and β can consistently achieve **>2% improvement in clean accuracy compared to one loss term alone** while keeping **ASR at 0%**.
>
> It seems to me that instead of using these combinations in an ad-hoc manner, it would have been useful to just use something like the SCRUB framework for unlearning [1], which minimizes the KL-div and CE on the retain set, and maximizes the KL-div on the forget set (which would be equivalent to the identified backdoored examples in your case). I do understand the authors' perspective that others would have complained about novelty in that case, but it seems to be scientifically better to reuse where possible, unless there is a good reason to deviate, which I am finding it difficult to see.
>
> **Response:**
> >We truly thank the reviewer for pointing out the SCRUB framework [1] as a potential alternative. We agree that SCRUB is an outstanding unlearning method designed to selectively remove unwanted knowledge while preserving desired knowledge. And its unlearning process can somehow be transferred to our approach as it minimizes the KL-divergence and CE loss on the retain set (to maintain performance on clean nodes), and (2) maximizes the KL-divergence on the forget set (to deliberately “erase” the targeted knowledge, analogous to poisoned nodes in our case).
>
> >We acknowledge that this design is theoretically appealing. However, we are **not sure whether SCRUB can be applied to various attack scenarios in graphs which may not fully align with SCRUB’s assumptions.** Our α–β combination strategy is designed to **flexibly weight different loss terms according to different scenarios**, and with certain settings, it can achieves substantial improvement in model performance. But a **fixed formulation** such as SCRUB may **not be equally effective across all settings**.
>
> >Due to time limits, we are not able to produce a GNN version of SCRUB in the current work, but we want to thank
>  you again for proposing this promising direction for future research and it could serve as a strong baseline for further comparison.
>
> >We highly appreciate your constructive feedback, which have greatly helped us improve the quality of our work. We look forward to exploring further with you soon.
>
> **Reference**
>
> Towards Unbounded Machine Unlearning, NeurIPS 2023. [1]

---

> > ### Comment · Reviewer_F71D · 2025-08-09
> > **Thank you for the updated results**
> >
> > Thank you for the additional results.
> >
> > I expected that there might be an issue with the implementation given that **ABL** unlearning has nothing different in it except GA. There is always a risk when asked to produce results on such short notice (my bad!). They align with what I would expect. I think the new results give a more holistic picture of what's happening.
> >
> > One thing I am still not convinced about is the loss terms. I would have been much happier if the authors had used their identification method coupled with a range of different baseline unlearning methods (something that they attempted to do in the rebuttal). I understand that there is a risk in that case to be perceived as less novel, but scientifically, it is more useful to build on existing literature.
> >
> > Given the effort the authors have put in during this rebuttal phase (I apologize if this was a little too much), I am happy to increase my score to advocate for an accept.
> >
> > However, for the next version of the paper, my sincere advice would be to get rid of their current unlearning algorithm, and do the experiments with existing well-known unlearning methods (specifically comparing the methods while splitting the identification and unlearning phases as the authors demonstrated during the rebuttal). My current assumption is that the experiments are not too expensive to run, which means that it is feasible to do these experiments again. This might be a more useful contribution in the long run compared to arguing about a particular adaptation of unlearning algorithms, which is not convincing enough.

---

> > > ### Author Response · Authors · 2025-08-09
> > > **Thank you for your constructive suggestions and updated score**
> > >
> > > Thank you very much for your inspiring feedback as well as for raising the score.
> > >
> > > We truly appreciate the time and effort you have invested in carefully reading our paper and providing beneficial suggestions.
> > > In the final version, we will refine our paper accordingly. We also believe this will allow us to better demonstrate the scientific value of our approach and provide a more comprehensive comparison within the existing literature.
> > >
> > > Thank you again for your valuable comments, which will certainly help us strengthen our work for the long term.

---

> > > > ### Comment · Reviewer_F71D · 2025-08-09
> > > > **Focusing on the core of the paper**
> > > >
> > > > The core contribution of the paper is the evaluation in a new domain. Hence, focusing on that core, it is useful to get scientific insights from it. Knowing your method works (which ultimately would be similar to other unlearning algorithms) is of little value. On the other hand, knowing which methods work, which methods don't work, and why they don't work is of tremendous value for future work. The separate evaluation for identification (which is a useful contribution as others methods aren't sufficient) and unlearning (which isn't particularly useful, since tons of useful methods already exist in that space) with evaluation of all major unlearning baselines would be a very useful point for the community.
> > > >
> > > > Hoping to see a more polished version of the paper coming out of this review process.

---

### Official Review · Reviewer_kDCE · 2025-07-02

**Clarity:** 3
**Significance:** 2
**Originality:** 1
**Rating:** 4
**Confidence:** 4

**Summary:**

The paper proposes LoSplit, a training-time defense against graph backdoor attacks by leveraging early-stage loss dynamics. It identifies poisoned target nodes via low z-score clustering under reverse cross-entropy loss and removes their influence through a decoupling-unlearning strategy. LoSplit achieves strong defense performance with minimal accuracy degradation.

**Questions:**

Could the authors clarify the reason for underlining specific results in Table 1?

**Ethical Concerns:**

["NO or VERY MINOR ethics concerns only"]

**Final Justification:**

Thank you for the clarification. Most of my concerns have been addressed, so I will raise my score to 4.

**Limitations:**

Yes

**Quality:**

3

**Strengths And Weaknesses:**

Strength

- LoSplit identifies poisoned nodes without relying on trigger patterns or label information, making it broadly applicable with reasonable transferability.
- The paper introduces an intriguing insight by leveraging RCE to distinguish backdoor target nodes more effectively.
- Experimental results shows that LoSplit significantly reduces ASR of various attack types (i.e. GTA, UGBA, DPGBA, SPEAR) across 4 graph datasets while maintaining clean accuracy.

Weakness

- The rationale for distinguishing target nodes from clean nodes based on differences in early-stage loss behavior has been extensively studied by backdoor defense methods such as ABL and DBD.
- The authors should more clearly explain how the “class-wide loss drift” phenomenon in graph-structured data renders defenses against graph backdoor attacks distinct from those in other domains, and specify which components of LosSplit are designed to address this challenge.
- Despite claiming evaluations on six datasets, the authors do not present results for two of them—Physics and Flickr.

---

> ### Author Rebuttal · Authors · 2025-07-29
>
> ## Rebuttal to Reviewer kDCE
>
> As both W1 and W2 concern the rationale and uniqueness of using early-stage loss behavior to separate target nodes in the context of graph-structured data, especially under the challenge of class-wide loss drift, we provide a unified response to W1 and W2.
>
> **W1. & W2.** The rationale for distinguishing target nodes from clean nodes based on differences in early-stage loss behavior has been extensively studied by backdoor defense methods such as ABL and DBD. The authors should more clearly explain how the “class-wide loss drift” phenomenon in graph-structured data renders defenses against graph backdoor attacks distinct from those in other domains, and specify which components of LosSplit are designed to address this challenge.
>
> #### **Response:**
> >We appreciate the concern regarding similarities to prior defenses such as **ABL** and **DBD**. While these methods also leverage **early-stage loss patterns**, they are primarily designed for **image-based backdoor attacks**, and their strategies can not directly transfer to **graphs** due to fundamental differences in both the **attack mechanisms** and the **loss behaviors**.
>
> >In **image domains**, each poisoned sample is a **standalone image**, analogous to a “whole graph” in graph domain with an embedded **trigger** typically in the form of black pixel, processed independently by the model. The trigger introduces **strong localized perturbations** that drive the model to quickly overfit these features, resulting in **consistently fast loss convergence** for poisoned samples in early training.
>
> >In contrast, **graph backdoor attacks** in node classification task are conducted over a **single connected graph**, where poisoned sample is a **node** embedded in the graph topology. The backdoored graph contains a large number of clean nodes and a small fraction of trigger nodes. Unlike in images, a trigger node does not influence the graph directly; Instead, it propagates **step-by-step** through **message passing** across n-hop neighboring nodes, but is constrained by **the limited number of hops**, which makes the loss behavior **unstable** as training epoch progresses (**W1.**).
>
> >As a result, unlike graph data, where the loss dynamics of nodes fluctuate significantly in early training, **image-based defenses** benefit from the **relatively stable loss distribution** when training epoch progresses (see Fig. 18 in Appendix I), which allows **fixed splitting ratios** or **global thresholds** (as used in **ABL**, **DBD**, and other methods in images) to work effectively. However, in **graph domains**, we observe a distinct phenomenon we call **class-wide loss drift**: the loss distribution of poisoned nodes **drifts unpredictably and overlaps with those of different clean classes**. This creates **blurred and class-dependent loss patterns** that vary significantly across epochs, as illustrated in **Fig. 1**. Consequently, the **global thresholds or static splitting ratio** used in image domain become ineffective in graph domain (**W2.**).
>
> >To address these **graph-specific challenges**, **LoSplit** introduces two key innovations. First, it performs **step-by-step** optimal split epoch selection using **inter-cluster divergence** (Eq. 7), rather than relying on a fixed heuristic early epoch like mainstream image approaches do. Second, it applies **class-wise z-score normalization** combined with **GMM clustering** to adaptively set split threshold and separate poisoned nodes based on **intra-class loss shifts and variance**. These techniques are designed specifically to **counteract the effects of class-wide loss drift** (**W2.**).
>
> >We summarize this distinction in the table below, which contrasts **LoSplit** with prior methods designed for **image domains**:
>
> **Table1:** Comparison of Defense Strategies Across Domains
> | **Domain** | **Method**  | **Defense Strategy* | **Splitting Ratio** | **Threshold** |
> |------------|-------------|------------------|----------------------|---------------|
> | Image      | ABL         | Unlearn          | Fixed                | –             |
> | Image      | DBD         | Suppress         | Adaptive             |    –          |
> | Image      | ASD         | Data Split       | Adaptive               | Global            |
> | Image      | HARVEY      | Data Split       | Adaptive             | Global         |
> | Graph      | **LoSplit** | Data Split       | **Adaptive**         | **Adaptive**  |
>
>
> **W3.** Despite claiming evaluations on six datasets, the authors do not present results for two of them—Physics and Flickr.
>
> #### **Response:**
> >Thank you for pointing this out. We sincerely apologize for the oversight. Our intention was to include results for all six datasets, including **Physics** and **Flickr**, but due to space limitations, we prioritized the most widely used benchmarks (**Cora**, **Citeseer**, **PubMed**, and **OGB-Arxiv**). Unfortunately, we neglected to remove the mention of Physics and Flickr from the main text after reformatting, which was our mistake.
>
> >We have completed the experiments on Physics and Flickr, and the results confirm the same trend: **LoSplit consistently reduces ASR to near zero while maintaining high clean accuracy**. The full results are provided below and will be included in Appendix of the revised paper.
>
> **Table2:** Defense Results on Physics and Flickr
> | Attack | Defense     | ASR↓ (Physics) | CA↑ (Physics) | ASR↓ (Flickr) | CA↑ (Flickr) |
> |--------|-------------|----------------|----------------|---------------|----------------|
> | GTA    | GCN         | 100.00         | 96.23          | 100.00        | 42.39          |
> |        | RobustGCN   | 100.00         | 94.98          | 99.89         | 40.44          |
> |        | GNNGuard    | 80.94          | 96.35          | 0.24          | 43.75          |
> |        | Prune       | 1.16           | 95.42          | 0.00          | 40.41          |
> |        | OD          | 0.00           | 96.46          | 0.00          | 41.47          |
> |        | ABL         | 100.00         | 96.25          | 0.00          | 40.80          |
> |        | RIGBD       | 100.00         | 96.43          | 0.00          | 43.98          |
> |        | **LoSplit** | **0.42**       | **96.58**      | **0.00**      | **44.29**      |
> | UGBA   | GCN         | 100.00         | 96.31          | 100.00        | 41.92          |
> |        | RobustGCN   | 99.98          | 95.23          | 90.25         | 40.34          |
> |        | GNNGuard    | 97.86          | 96.06          | 99.07         | 46.80          |
> |        | Prune       | 95.73          | 95.16          | 90.23         | 40.45          |
> |        | OD          | 0.00           | 96.20          | 0.00          | 41.25          |
> |        | ABL         | 1.93           | 95.19          | 0.00          | 36.85          |
> |        | RIGBD       | *0.56*          | **96.38**      | 0.00          | 43.08          |
> |        | **LoSplit** | **0.00**       | 96.35          | **0.00**      | **43.25**      |
> | DPGBA  | GCN         | 94.35          | 96.25          | 99.57         | 45.19          |
> |        | RobustGCN   | 94.44          | 96.35          | 95.61         | 40.95          |
> |        | GNNGuard    | 95.59          | 95.74          | 4.50          | 45.46          |
> |        | Prune       | 1.61           | 96.23          | 0.00          | 40.62          |
> |        | OD          | 94.52          | 96.25          | 98.56         | 42.59          |
> |        | ABL         | 81.85          | 93.30          | 50.16         | 40.26          |
> |        | RIGBD       | 0.98           | 96.27          | 0.00          | 45.12          |
> |        | **LoSplit** | **0.34**       | **96.46**      | **0.00**      | **45.32**      |
> | SPEAR  | GCN         | 95.36          | 96.27          | 100.00        | 45.56          |
> |        | RobustGCN   | 90.91          | 96.30          | 98.91         | 40.43          |
> |        | GNNGuard    | 63.48          | 96.14          | 71.84         | 44.64          |
> |        | Prune       | 96.78          | 96.15          | 100.00        | 40.52          |
> |        | OD          | 53.92          | 96.22          | 41.59         | 41.48          |
> |        | ABL         | 11.56          | 94.69          | 100.00        | 40.59          |
> |        | RIGBD       | 88.03          | 96.35          | 100.00        | 44.24          |
> |        | **LoSplit** | **0.00**       | **96.42**      | **0.00**      | **45.69**      |
>
>
>  **Q.** Could the authors clarify the reason for underlining specific results in Table 1?
> #### **Response:**
> >We sincerely apologize for the confusion caused by the underlining results in Table 1 and the lack of clear statement. We underlined those results to highlight an important observation: even when certain baselines achieve their best clean accuracy (ACC), their attack success rate (ASR) remains relatively high and vice versa, indicating insufficient robustness. This contrast emphasizes that achieving high ACC alone is not enough; an effective defense must maintain both high ACC and low ASR simultaneously.

---

> ### Comment · Reviewer_kDCE · 2025-08-06
>
> Thank you for the clarification. Most of my concerns have been addressed, so I will raise my score to 4.

---

> ### Author Response · Authors · 2025-08-06
> **Official Comment by Authors**
>
> We are truly grateful for your reconsideration. We’re glad to hear that our responses addressed your concerns and sincerely appreciate your updated assessment.

---

### Official Review · Reviewer_5s8N · 2025-07-18

**Clarity:** 3
**Significance:** 3
**Originality:** 3
**Rating:** 4
**Confidence:** 3

**Summary:**

This paper proposes LoSplit, a novel training-time defense framework targeting graph backdoor attacks, particularly those based on feature perturbations rather than structural triggers. The authors identify a key observation: target nodes poisoned by backdoor triggers tend to converge earlier in training, leading to an early-stage loss divergence, especially under reverse cross-entropy (RCE) loss. Leveraging this, LoSplit dynamically identifies a training epoch with maximal loss separability, clusters nodes within the target label using Gaussian Mixture Models (GMMs), and applies a Decoupling-Unlearning strategy to de-associate target nodes from the malicious label. Extensive experiments on multiple benchmark datasets (Cora, Citeseer, Pubmed, OGB-arxiv) and attack methods (GTA, UGBA, DPGBA, SPEAR) show that LoSplit achieves state-of-the-art defense performance, significantly lowering attack success rates (ASR) while preserving clean accuracy.

**Questions:**

## Detailed Comments and Questions

1. **Adaptive Robustness Concerns**
   The proposed method hinges on an empirical observation: target nodes tend to converge faster under CE/RCE loss. However, this loss-based separation is fragile—adaptive attackers could explicitly regularize poisoned node loss to mimic clean behavior.
   - Have the authors considered evaluating the method under such adaptive scenarios?
   - Adding a section that simulates these adaptive attacks (e.g., using loss matching objectives during poisoning) would substantially strengthen the claims of robustness.

2. **Theoretical Limitation**
   The theorem in the paper demonstrates that a maximum in loss separation exists. However, this is insufficient for establishing robustness.
   - Can the authors provide a bound or probabilistic guarantee on the separation margin between clean and poisoned nodes?
   - Without this, it's unclear whether the GMM clustering will remain effective across different datasets or attack configurations.

3. **Behavior on Clean Datasets**
   There is no analysis of the method’s behavior when the dataset is not backdoored.
   - What happens when the method is applied to a clean dataset?
   - Does it mistakenly reassign or discard normal nodes?
   - An evaluation on this scenario is critical to understand the false positive rate and the cost of deploying the defense in practice.

4. **Random Label Reassignment in Section 4.3**
   The paper mentions that poisoned nodes are assigned random new labels. This seems arbitrary.
   - What is the underlying motivation for this choice?
   - Would discarding these nodes or reinitializing their features be more effective?
   - An ablation study comparing different handling strategies would provide clarity on this decision.

**Ethical Concerns:**

["NO or VERY MINOR ethics concerns only"]

**Final Justification:**

The authors provided thorough and thoughtful responses to my concerns. They addressed the risk of false positives on clean datasets with empirical evidence, clarified the rationale behind random label reassignment with an ablation study, and pointed to adaptive thresholding as a principled way to separate poisoned nodes. While robustness under adaptive attacks remains an open issue, I believe the method is well-motivated, empirically strong, and offers valuable insights. Based on the rebuttal, I am increasing my score to reflect a positive view of the paper’s contributions.

**Limitations:**

yes

**Quality:**

2

**Strengths And Weaknesses:**

### Strengths

- The paper proposes a novel training-time defense mechanism that identifies and mitigates backdoor-poisoned nodes early in the learning process. This approach focuses on leveraging loss dynamics and clustering to isolate potentially malicious nodes.
- The proposed method demonstrates promising empirical results across multiple benchmark datasets and various attack methods, suggesting practical effectiveness in the evaluated settings.

### Weaknesses
- The defense is likely brittle under adaptive attacks. Its core assumption—that poisoned nodes exhibit faster loss convergence under CE or RCE—is easily circumvented by attackers who align poisoned node loss curves with clean nodes during training.
- The theoretical contribution is limited. The current theorem only establishes the existence of a separation point but does not quantify the separability margin or offer robustness guarantees.
- The method has not been tested on clean (unpoisoned) datasets. This raises concerns about false positives and unintended degradation of clean accuracy.
- The rationale behind certain design choices, such as random label reassignment for detected poisoned nodes, is unclear and lacks empirical justification.

---

> ### Author Rebuttal · Authors · 2025-07-29
>
> ## Rebuttal to Reviewer 5s8N
> We would like to thank the reviewer for the constructive feedback and for highlighting important areas of discussion. Due to the fact that your Questions are an extension of the Weaknesses, we provide a unified response to both below.
>
> **W1. & Q1.**  **The defense is likely brittle under adaptive attacks.**  Its core assumption—that poisoned nodes exhibit faster loss convergence under CE or RCE—is easily circumvented by attackers who align poisoned node loss curves with clean nodes during training.   The proposed method hinges on an empirical observation: target nodes tend to converge faster under CE/RCE loss. However, this loss-based separation is fragile—adaptive attackers could explicitly regularize poisoned node loss to mimic clean behavior.  Have the authors considered evaluating the method under such adaptive scenarios? Adding a section that simulates these adaptive attacks (e.g., using loss matching objectives during poisoning) would substantially strengthen the claims of robustness.
>
>
> #### **Response:**
> >We appreciate this concern and agree that adaptive attacks are an important consideration. **As a defense-side work**, our goal is to counter known and practical attack scenarios. Our method targets widely adopted graph backdoor attacks (GTA, UGBA, DPGBA, SPEAR), where the rapid early-stage convergence of poisoned (target) nodes is a **consistent and empirically observable behavior (see Sec. 4.1, Fig. 2)**. We also **provide theoretical support** for this observation in Appendix C.2. To the best of our knowledge, such a threat model has not been considered in existing graph backdoor attack literature. That said, we respectfully agree that this is an important direction. We thank the reviewer for highlighting it, and it is an interesting investigation for future work regarding such attack and corresponding defense.
>
>  **W2. & Q2.** **The theoretical contribution is limited.**
> The current theorem only establishes the existence of a separation point but does not quantify the separability margin or offer robustness guarantees.  The theorem in the paper demonstrates that a maximum in loss separation exists. However, this is insufficient for establishing robustness.  Can the authors provide a bound or probabilistic guarantee on the separation margin between clean and poisoned nodes?  Without this, it's unclear whether the GMM clustering will remain effective across different datasets or attack configurations.
>
> #### **Response:**
> >We respectfully agree that robustness guarantees are important for the reliability of any defense. However, our paper **does define the bound on the separation margin explicitly in Eq. (6)** through the adaptive threshold $\tau^{(t)}$. This threshold is not heuristic: it is calculated as the empirical midpoint between two dynamically identified clusters: clean and potential target nodes within the same target class at epoch $t$. More specifically, this adaptive threshold defines a **lower bound** for clean nodes and an **upper bound** for poisoned nodes, and the margin is directly computable through loss clustering. Unlike fixed or manually tuned thresholds, $\tau^{(t)}$ adjusts automatically to dataset-specific and attack-specific scenarios. In practice, this adaptive threshold consistently achieves robust performance across datasets and attacks (see Table 2), suggesting its effectiveness even **without explicit probability assumptions**. We will revise Section 4.2 to make it clear that Eq. (6) defines an explicit, statistically grounded bound that offers practical robustness without relying on strong probability assumptions.
>
> **W3. & Q3.** **The method has not been tested on clean (unpoisoned) datasets.**
> This raises concerns about false positives and unintended degradation of clean accuracy.  There is no analysis of the method’s behavior when the dataset is not backdoored.  What happens when the method is applied to a clean dataset?  Does it mistakenly reassign or discard normal nodes? An evaluation on this scenario is critical to understand the false positive rate and the cost of deploying the defense in practice.
>
> #### **Response:**
> >Thank you for raising this significant question. LoSplit **does not mistakenly reassign or discard any nodes when applied to a clean dataset**, because our method relies on the **difference in early-stage loss drop rate** to distinguish poisoned nodes from clean ones. In backdoored graphs, poisoned nodes typically converge faster due to shortcut learning. However, in a **clean dataset**, such difference does **not exist**, thus all nodes follow similar loss trajectories. Consequently, the dynamic splitting (Eq. (6)) does not identify any target-like cluster, and **nearly no nodes are marked for reassignment**. This is by our algorithm design: when our method detects no **bimodal loss distribution** within the target class—i.e., no clear separation between clean and poisoned nodes, it automatically sets the threshold to a very small value (e.g., 1e-5), effectively preventing any nodes from being reassigned.
>
> >To verify this, we compare GCN (Without Defense) and LoSplit (With Defense) on clean graph without injecting triggers. The tables below  reports the **Clean Accuracy (CA%)** and **False Positive Rate (FPR%)**:
>
> ---
> **Table 1.** Clean accuracy (CA, %) comparison between GCN and LoSplit on clean datasets.  This table reports the clean accuracy of GCN trained with and without the LoSplit defense on clean graph.
> | Model                | Cora | Citeseer | Pubmed |
> |----------------------|------|----------|--------|
> | GCN (No Defense)     | 83.70 | 74.70    | 85.18  |
> | LoSplit (Ours)       | 83.33 | 74.39    | 85.03  |
>
> ---
>
> **Table 2.** False positive rate (FPR, %) of LoSplit on clean datasets.  FPR is defined as the percentage of clean nodes mistakenly identified as poisoned.
> | Dataset   | Cora | Citeseer | Pubmed |
> |-----------|------|----------|--------|
> | FPR (%)   | 0.18 | 1.05     | 0.48   |
>
> >**Key Takeaway:** Across all datasets, we observe that our LoSplit achieves comparable clean accuracy to no defense settings and near zero FPR, confirming that LoSplit rarely mistakenly choose clean nodes as poisoned ones. We will add this analysis in the Appendix of revised paper under *Behavior Under Clean Graphs* .
>
> **W4. & Q4.** **Random Label Reassignment in Section 4.3**  The rationale behind certain design choices, such as random label reassignment for detected poisoned nodes, is unclear and lacks empirical justification. The paper mentions that poisoned nodes are assigned random new labels. This seems arbitrary. What is the underlying motivation for this choice?  Would discarding these nodes or reinitializing their features be more effective? An ablation study comparing different handling strategies would provide clarity on this decision.
>
> #### **Response:**
> >We appreciate the reviewer’s thoughtful comment. The **underlying rationale and motivation** for random label reassignment is to break the **shortcut learning between target nodes and the malicious labels.**  Reassignment  is  a **kind of unlearning**. Backdoor triggers strongly bind the target class to target nodes. If we retain their malicious label, the model continues to reinforce this shortcut during retraining.  In our approach, random labels are reassigned at every training iteration, so each target node is exposed to a different random label in each epoch. This dynamic labeling acts as a diverse perturbation during training, **effectively disrupting the shortcut learning process for target nodes and preventing target nodes from overfitting to any specific incorrect label.**
>
> >To validate this, we conducted an **ablation study** comparing different strategies mentioned in your feedback, we use PubMed dataset under SPEAR and UGBA attack:
>
> ---
>
> **Table 3.** Effect of different retraining strategies on attack success rate (ASR, %) and clean accuracy (ACC, %) under SPEAR and UGBA attacks.
> | Strategy                 | ASR ↓ (SPEAR) | ACC ↑ (SPEAR) | ASR ↓ (UGBA) | ACC ↑ (UGBA) |
> |--------------------------|----------------|----------------|----------------|----------------|
> | Random Label (Ours)      | **0.00**        | **84.93**        | **0.19**        | **85.18**        |
> | Node Removal             | 0.33           | 84.53           | 89.06           | 85.03           |
> | Feature Reinitialization | 0.00           | 84.74           | 90.09           | 84.83           |
>
> >**Key Takeaway:** Random reassignment achieves the lowest ASR while maintaining highest ACC, outperforming other strategies. We will add this table to Appendix and clarify the rationale in Section 4.3. **Note that on UGBA attack which craft a backdoor trigger that mimics the neighbor of target nodes. Even if we remove the target nodes from the training set, those attack can still lead to a high ASR.**
>
> >**Why Not Node Removal?**
>
> >**Removing target nodes** entirely from the training graph may seem like a fair solution, but it raises two problems: **(1)**  Removing nodes breaks neighborhood connectivity, which may degrade generalization. **(2)** It is ineffective against attacks in which triggers can affect the neighborhood of target nodes. Some attacks like UGBA **craft a backdoor trigger that mimics the neighbor of target nodes.** Even if the target nodes are removed, their influence still exists. This explains why node removal still results in **high ASR (89.06%) under UGBA**.
>
> >**Why Not Feature Reinitialization?**
>
> >**Reinitializing node features** tries to recover the feature of target nodes, but it assumes that features alone are responsible for the attack, which fails under **structural-based attack** (e.g., UGBA). The reinitialized features can still influence UGBA, since they retain the correlation between target nodes and malicious label, the model can still pick up on weak indirect correlations. Hence, ASR remains high (90.09%).

---

> > ### Comment · Reviewer_5s8N · 2025-08-06
> >
> > The authors provided thorough and thoughtful responses to my concerns. They addressed the risk of false positives on clean datasets with empirical evidence, clarified the rationale behind random label reassignment with an ablation study, and pointed to adaptive thresholding as a principled way to separate poisoned nodes. While robustness under adaptive attacks remains an open issue, I believe the method is well-motivated, empirically strong, and offers valuable insights. Based on the rebuttal, I am increasing my score to reflect a positive view of the paper’s contributions.

---

> ### Author Response · Authors · 2025-08-06
> **Official Comment by Authors**
>
> We sincerely appreciate your thoughtful feedback and encouraging response. Your acknowledgement of our empirical analysis and design is truly motivating and strengthens our confidence in improving this work in the future.

---

### Note · Authors · 2025-08-12

We truly appreciate all the reviewers and the ACs for their enlightening feedback!

 **All reviewers consistently hold a positive view of our work** and **three of them (#5s8N, #kDCE, #F71D) have committed to improving their scores** and **believe that our paper deserves acceptance.** They share a common recognition of our main strengths and contributions, which can be summarized below:

1. **First training-time defense in graph:** LoSplit is the first training-time defense for graph backdoor attacks in node classification task, which is important and unexplored. [#5s8N, #kDCE, #F71D, #e9Mr]

2. **Strong empirical results:**  LoSplit significantly reduces the ASR of diverse attack types across various widely used graph datasets while maintaining high clean accuracy, and it achieves substantial improvements over the state-of-the-art method RIGBD. [#5s8N, #kDCE, #F71D, #e9Mr]

3. **New perspective:** LoSplit enables more accurate identification of poisoned nodes by amplifying early loss dynamics via RCE term,  thereby avoiding reliance on graph topology or outlier assumptions, particularly effective for the stealthiest feature-perturbation attack SPEAR. [#kDCE, #e9Mr]

Nonetheless, a few concerns and questions were raised (e.g., robustness against specific attack types, rationale behind Decoupling-Unlearning strategy, and additional ablation studies), **which we have addressed thoroughly during our rebuttal.**

**We are grateful that all reviewers have acknowledged our clarifications, reconsidered their opinions regarding the contributions and impact of our paper, and believed our paper can be accepted.**

We would like to thank all the reviewers and the ACs again for their time and effort.

---

### Decision · Program_Chairs · 2025-09-17

**Decision:**

Accept (poster)

**Comment:**

The reviewers unanimously recommend acceptance of the paper with varying degrees of strength. I agree with their assessment and am happy to recommend acceptance of this paper for publication at the NeurIPS conference.

The reviews and the rebuttal have given rise to several interesting points and results that I encourage the authors to include in their revised manuscript. This includes the study of the performance on clean, i.e., unattacked, graphs. I think this study should be extended beyond Cora, CiteSeer and Pubmed to also the remaining three datasets you consider. I furthermore encourage the inclusion of the ablation studies on the random label assignment that you provided, as well as the results on the Physics and Flickr datasets.